



# Measurement report: Production and loss of atmospheric formaldehyde at a suburban site of Shanghai in summertime

**Yizhen Wu[1], Juntao Huo[2], Gan Yang[1], Yuwei Wang[1], Lihong Wang[1], Shijian Wu[2], Lei Yao[1], Qingyan Fu[2], Lin Wang[1,3,4,5,6]**

[1] Shanghai Key Laboratory of Atmospheric Particle Pollution and Prevention (LAP[3]), Department of Environmental Science & Engineering, Jiangwan Campus, Fudan University, Shanghai 200438, China

[2] Shanghai Environmental Monitoring Center, Shanghai 200030, China

[3] Collaborative Innovation Center of Climate Change, Nanjing, 210023, China

[4] Shanghai Institute of Pollution Control and Ecological Security, Shanghai 200092, China

[5] IRDR International Center of Excellence on Risk Interconnectivity and Governance on Weather/Climate Extremes Impact and Public Health, Fudan University

[6] National Observations and Research Station for Wetland Ecosystems of the Yangtze Estuary, Shanghai, China

*Correspondence to*: Qingyan Fu (qingyanf@sheemc.cn) and Lin Wang (lin_wang@fudan.edu.cn)

**Abstract.** Formaldehyde (HCHO) is an important trace gas that affects the abundance of $HO_2$ radicals and ozone, leads to complex photochemical processes, and yields a variety of secondary atmospheric pollutants. In a 2021 summer campaign at the Dianshan Lake (DSL) Air Quality Monitoring Supersite in a suburban area of Shanghai, China, we measured atmospheric formaldehyde (HCHO) by a commercial Aero-Laser formaldehyde monitor, methane, and a range of non-methane hydrocarbons (NMHCs). Ambient HCHO showed a significant diurnal cycle with an average concentration of $2.2\pm1.8$ ppbv (parts per billion by volume). Secondary production of HCHO was estimated to be approximately 70% according to $HCHO/NO_x$. The average secondary HCHO production rate was estimated to be 0.73 ppbv $h^{-1}$, with a dominant contribution from reactions between alkenes and OH radicals (66.3%), followed by OH radical-initiated reactions with alkanes and aromatics (together 19.0%), OH radical-initiated reactions with OVOCs (8.7%), and ozonolysis of alkenes (6.0%). An overall HCHO loss, including HCHO photolysis, reactions with OH radicals, and dry deposition, was estimated to be 0.49ppbv $h^{-1}$. The net HCHO production was in good agreements with the observed rate of HCHO concentration change throughout the sunny days, indicating that HCHO was approximately produced by oxidation of the 24 hydrocarbons we took into account at the DSL site during the campaign, where primary emissions and transport processes can likely be excluded. Our results suggest the important role of secondary pollution at the suburb of Shanghai, where alkenes are likely key precursors for HCHO.

## 1 Introduction

Formaldehyde (HCHO) is the most abundant carbonyl in the troposphere, which is an intermediate product from the oxidation of various volatile organic compounds (VOCs) and plays an important role in various photochemical processes as both a source and a sink of free radicals (Wittrock et al., 2006). Photolysis of HCHO produces $HO_2$ radicals that can be





subsequently converted to OH radicals, altering the oxidative capacity of the atmosphere (Mahajan et al., 2010; Tan et al., 2019a). HCHO can also contribute to ozone ($O_3$) formation, and their link is one of the most popular topics of atmospheric research (Hong et al., 2022; Liu et al., 2007; Pavel et al., 2021). Furthermore, HCHO can play a crucial catalytic role in the

formation of particulate matters via in-cloud processing pathways to form hydroxymethyl hydroperoxide (HMHP) or hydroxymethanesulfonate (HMS) (Dovrou et al., 2022). Besides, HCHO has adverse health effects on humans and animals, possibly causing respiratory and cardiovascular diseases (Zhu et al., 2017).

Direct emission of HCHO can be attributed to anthropogenic activities, such as vehicle exhaust, industrial emissions, and fossil fuel combustion, and thus primary HCHO is closely related to anthropogenic pollutants such as CO, $NO_x$ and traffic-

related black carbon (BC) (Zhang et al., 2013; Dutta et al., 2010; Possanzini et al., 2002; Lin et al., 2012). Observations in city centers usually showed important HCHO sources of direct anthropogenic emissions (Dutta et al., 2010; Possanzini et al., 2002; Lin et al., 2012). On the other hand, previous studies indicate that secondary production of HCHO plays an important role in remote areas, which comes from complex oxidation processes of a wide range of VOCs by ubiquitous atmospheric oxidants like OH radicals and $O_3$ (Lin et al., 2012; Anderson et al., 2017; Nussbaumer et al., 2021). Especially, in summertime, increases

in temperature are always linked with higher solar radiation and larger biogenic isoprene emission, which have been proved to jointly contribute to the higher abundance of HCHO (Choi et al., 2010; Sumner et al., 2001; Wang et al., 2017). Due to the short lifetime of HCHO, primary emissions and transport processes are important in source regions but can be mostly neglected in remote locations. On the other hand, secondary HCHO production paths are more diverse and complex, and thus they are difficult to be quantified.

In previous studies, several methods were used to separate the primary and secondary sources of HCHO, including the emission ratios of HCHO-to-tracers from primary sources (Possanzini et al., 2002; Lin et al., 2012), the photochemical age-based parameterization (de Gouw et al., 2005, 2018; Wang et al., 2017), receptor models such as the positive matrix factorization (PMF) model and the principal component analysis (PCA) model (Wang et al., 2017; Chen et al., 2014), and the multi-linear regression method based on ambient measurements of HCHO and various tracers that can represent primary and

secondary sources, respectively (Su et al., 2019; Zhang et al., 2021b). These methods just provide a simple estimation on the ratios of different HCHO sources. Nevertheless, contributions of different precursors are the key to understand and prevent HCHO production, but a qualitative and quantitative understanding on the HCHO production and loss is still incomplete.

HCHO sources have been previously studied in China (Xing et al., 2020; Su et al., 2019; Wang et al., 2017; Sun et al., 2021; Zhang et al., 2021b), in which general fractions of primary and secondary HCHO were provided and important roles of

secondary HCHO sources were suggested. Primary and secondary contributions to ambient HCHO were separated using a multiple linear regression based on HCHO observed by Ozone Mapping and Profiler Suite (OMPS) in the Yangtze River Delta

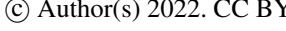

in China, suggesting that secondary formation, especially photochemical production, played crucial roles (Su et al., 2019; Zhang et al., 2021b). A more recent study used the GEOS-Chem model to stimulate the concentrations of HCHO over Hefei Province in summer, indicating that oxidations of both methane and nonmethane VOCs dominated the HCHO production with

a contribution of 43.27% and 56.73%, respectively (Sun et al., 2021). To the best of our knowledge, little has been done to quantify the contribution of various specific precursors to HCHO production in China.

On the other hand, studies that have estimated the secondary HCHO production and discriminated the contribution of different secondary HCHO precursors in North America and Europe still found missing production terms (Lin et al., 2012; Sumner et al., 2001). For example, missing HCHO production rates of 1.1-1.6 ppbv h$^{-1}$ were reported for a remote site in

America, which was nearly double of the calculated secondary production rates (Choi et al., 2010). Moreover, very few studies have evaluated the HCHO budget by comparing the net production of HCHO (P(HCHO)-L(HCHO)) and the observed rate of HCHO concentration change, and even for those with such attempts, discrepancies remained (Sumner et al., 2001; Zhang et al., 2021a). Therefore, an estimation of secondary HCHO production based on a comprehensive observation of HCHO precursors, as well as a comprehensive understanding on the formation and loss of HCHO to fill the gap in the HCHO budget

calculation, is urgently needed to reveal the key precursors to HCHO formation, especially in eastern China where photochemical pollution is getting severe in recent years.

In this study, we measured atmospheric HCHO concentrations, using a commercial Aero-Laser formaldehyde monitor, from June 10 to July 4, 2021 at the Dianshan Lake (DSL) Air Quality Monitoring Supersite, a suburban area of Shanghai. The secondary HCHO production rates were estimated based on parallel measurements of O$_3$, photolysis frequencies, and 24 VOCs

whose photochemical reactions lead to formation of HCHO. The loss of HCHO including HCHO photolysis, reactions with OH radicals, and dry deposition was estimated. Also, characteristics of secondary HCHO production between the sunny period and the cloudy and rainy period were compared. Lastly, the net production of HCHO was compared with the observed rate of HCHO concentration change.

## 2 Methodology

### 2.1 Measurement site

A comprehensive campaign was carried out at the DSL Air Quality Monitoring Supersite in Shanghai, China (31.10°N, 120.98°E), covering June 10 to July 4, 2021. The station is located in Qingpu District in the western suburb of Shanghai, which is about 37 km away from the city center, and surrounded by Dianshan Lake and several villages with a small quantity of residents (Figure 1). There is a highroad (G318) about 0.4 km southeast of the sampling site. Since it is located at the junction



of Shanghai, Zhejiang Province and Jiangsu Province, all of which are well developed areas with large populations, this super

site is often affected by regional transport and suffers from photochemical pollution episodes (Yang et al., 2022).

**2.2 Measurements of VOCs**

    Atmospheric HCHO was measured by a commercial Aero-Laser formaldehyde monitor (Aerolaser GmbH., model AL4021)

with a detection limit (DL) of 100 pptv (parts per trillion by volume). In this instrument, gaseous HCHO is firstly absorbed by

sulfuric acid (0.3%) and transferred into liquid phase, and then the aqueous HCHO reacts with acetylacetone and ammonia to

produce 3,5-diacetyl-1,4-dihydrolutidine (DDL), which can be excited by a laser at 400 nm and the fluorescence at 510 nm is

used to quantify HCHO. Liquid formaldehyde standards of 80 μg $L^{-1}$ were used to calibrate the instrument weekly. HCHO

was measured at a time resolution of 5 min but the obtained data was processed to those at 1 h intervals to match the VOC

data from an online gas chromatograph equipped with mass spectrometry and flame ionization detection (GC-MS/FID).

Atmospheric methane was measured with GC-FID (ChromaTHC, model C24022). Methane was separated from ambient

non-methane hydrocarbons (NMHCs), since NMHCs were concentrated in a packed stainless-steel column filled with Porapak

Q, 50/80 Mesh, whereas methane was able to freely pass the column and to be detected by a hydrogen ion flame detector.

Methane standards of 2 ppmv (parts per million by volume) were used to perform instrument calibration every week, and those

at five concentration levels (0, 1, 2, 3, 4 ppmv) were used to quantify the concentration of ambient methane. The concentration

of methane was detected at a time resolution of 5 min but averaged to those at 1 h intervals to match the GC-MS/FID data.

    Measurements of Photochemical Assessment Monitoring Station (PAMS) VOC compounds were performed with GC-

MS/FID (TH-PKU 300B) with a time resolution of 1 hour, given the time for heating and cooling of the GC oven (Li et al.,

2015). Ambient concentrations of 2-methyl pentane were always below the detection limit of this compound in the instrument,

and thus the corresponding data was not included. Ambient air was sampled into the system, where C2-C5 hydrocarbons were

separated by a Porous Layer Open Tubular column (PLOT) (Agilent Technologies Inc.) and measured by the FID channel,

and the other hydrocarbons (C6-C12) were preconcentrated by a semi-polar column (DB-624, Agilent Technologies Inc.) and

detected using a quadrupole MS detector. Weekly calibration was conducted by a built-in auto-calibration system, using

gaseous bromochloromethane, 1,2-difluorobenzene, chlorobenzene-d5, and bromofluorobenzene in high purity $N_2$, each of

which is of 4 ppbv. Mixtures of 57 PAMS compounds at five concentration levels (0, 1, 2, 4, 8 ppbv) were used to quantify

the ambient concentrations of these species.

    Isoprene, alpha-pinene, methyl vinyl ketone (MVK) and methacrolein (MACR) were detected by a two-channel gas

chromatograph with electron ionization time-of-flight mass spectrometer (GC-EI-TOF-MS) system. The GC-EI-TOF-MS

system consists of three main components: (1) a thermal desorption pre-concentrator (TDPC) (Aerodyne Research Inc.) for

sample collection, (2) a gas chromatograph (GC) (Aerodyne Research Inc.) for sample separation, and (3) an electron





ionization time-of-flight mass spectrometer (EI-TOF-MS) (Tofwerk AG, model EI-003) for sample detection (Gilman et al., 2013; Obersteiner et al., 2016; Claflin et al., 2021). The TDPC employed for this campaign relied upon a two-stage adsorbent trap for preconcentration of analytes, allowing a large sample volume of 1 L of ambient air. During this campaign, the sampling flow of each trap was about 800 sccm (standard cubic centimeter per minute). The ARI GC was configured as a two-channel system to expand the volatility range, with a 30 m Rxi-624 analytical column (Restek, 0.25 mm ID, 1.4 μm film thickness) for channel-1 and a 30 m MXT-WAX analytical column (Restek, 0.25 mm ID, 0.25 μm film thickness) for channel-2 installed in the spindle. Isoprene, alpha-pinene, MVK and MACR were all analyzed on channel-1, which is optimized for C5-C12 hydrocarbons, along with oxygen-, nitrogen-, halogen-, and sulfur-containing VOCs. During the sample acquisition, $H_2O$, $CO_2$, $O_2$ and $O_3$ were removed prior to cryogenically trapping of VOCs, i.e., $H_2O$ was captured by a cryogenic de-watering cold trap, $CO_2$ and $O_2$ were purged by the carrier gas, and $O_3$ was absorbed by an oxidant trap. Automated gas-phase calibration of GC-EI-TOF-MS was conducted every 20 hours with a standard gas cylinder (Apel-Riemer Environmental Inc.), including MVK, furan, propanal, methyl tert-butyl ether, butanal, ethyl acetate, toluene, octane, m-xylene, o-xylene, naphthalene, 1-methylnaphthalene, decamethylcyclopentasiloxane, and limonene. In order to guarantee the stability of the instrument, background signal was determined at the same frequency. MVK was calibrated during the campaign with the automated gas-phase calibration as mentioned above. Isoprene and alpha-pinene were calibrated after the campaign using another calibrant cylinder (Weichuang Standard Reference Gas Analytical Technology Co., Ltd.), where a 5-point calibration was conducted and each point was repeated for 3 times. As for MACR, calibration was conducted after the campaign with a liquid calibration system (LCS, Tofwerk AG). Briefly, liquid MACR were mixed with methanol that was used as a solvent, and injected steadily into a stream of 2 slpm (standard liters per minute) ultra-high-purity $N_2$ with a high precision syringe pump, part of which (around 198 sccm) was guided into the GC-EI-TOF-MS. The sampling time of GC-EI-TOF-MS was adjusted to obtain a 4-point calibration curve with 3 duplicates for each point. The consistency between the liquid calibration and the calibrant cylinder calibration was confirmed by conducting both calibration methods for m-xylene, o-xylene, 1,2,4-TMB, and benzene. The sensitivity of the GC-EI-TOF-MS during and after the campaign was calibrated and normalized using the first standard gas cylinder (Apel-Riemer Environmental Inc.). Concentrations of isoprene, alpha-pinene, MVK and MACR were measured with a time resolution of 30 min, which were later averaged into those at 1 h intervals to match the GC-MS/FID data.

Isoprene was detected by both GC-MS/FID (DL of 0.04 ppbv for isoprene) and GC-EI-TOF-MS (DL of 0.77 pptv), and the corresponding inter-comparison is shown in Figure S1. Daytime isoprene concentrations showed excellent agreements between two systems (Figure S1a), whereas at nighttime, GC-MS/FID usually showed higher results than GC-EI-TOF-MS when concentrations of isoprene declined to low values, as shown in Figure S1b. The reason for the discrepancy at night is unknown. Considering the lower detection limit and higher accuracy of GC-EI-TOF-MS, isoprene measured by GC-EI-TOF-

MS will be used in the following discussion. The good correlation between daytime isoprene concentrations suggests minor uncertainties of our daytime isoprene concentrations. Though there was a discrepancy between nighttime isoprene concentrations, the concentrations of measured isoprene and estimated OH radicals at night were so low that they hardly affect our results of calculated secondary HCHO production rates.

**2.3 Measurements of other pollutants and meteorological parameters**

$O_3$ was measured with an ultraviolet photometric analyzer (Thermo Environmental Instruments, TEI Inc., Model 49i) with a DL of 1 ppbv (at a time interval of 10 s). NO and $NO_2$ mixing ratios were determined using a chemiluminescent analyzer (TEI, Model 42i), DL for which is 0.40 ppbv (at a time interval of 60 s).

Photolysis frequencies of HCHO, $NO_2$, $NO_3$, $O^1D$, HONO, $H_2O_2$ were determined with an Ultra-fast CCD-Detector Spectrometer (Metcon, UF-CCD), with a time resolution of 1 min. The instrument consists of an optical receiver, a Charge-

Coupled detector (CCD) and a cooling control box for the detector, and covers spectral band range from 280 ~ 650 nm. The photolysis frequencies can be calculated automatically from the measured spectra and the calibration factor, which is obtained from the calibration using NIST/PTB 1000W halogen lamps.

Temperature (T), relative humidity (RH), atmospheric pressure (P), wind speed (WS), wind direction (WD) and rainfall were captured by an automatic and commercial weather monitoring station (Vaisala AWS310). Planetary boundary layer (PBL)

depth was recorded by a ceilometer (Vaisala, CL31). Atmospheric parameters were measured at a time resolution of 5 min.

All trace gases, meteorological parameters and photolysis frequencies were then processed to those at 1 h intervals to match the GC-MS/FID data.

More details about the detection limit and accuracy of trace gases, VOCs, photolysis frequencies and PBL can be found in Table S2. Total uncertainties of calculated HCHO production and loss rates were estimated by applying the root square

propagation of corresponding uncertainties of quantities used for the calculation.

**3 Results**

**3.1 Overview of the campaign**

This campaign was carried out from June 10 to July 4, 2021 at the DSL site. The sampling period coincided with the Plum Rain Season, a typical east Asian rainy period that features several weeks of wet days as well as high temperatures and usually

starts in June, causing a significant weather variation during the campaign.

Figure 2 shows the time profiles of HCHO, $O_x$ (=$NO_2$+$O_3$), $O_3$, $NO_x$, photolysis frequencies of $O^1D$ ($J(O^1D)$) and meteorological parameters during the campaign. Datapoints are sometimes missing in the case of instrument routine



calibrations and a couple of instrument failure. In addition, Table S1 summarizes the averages and the $10^{th}$ and $90^{th}$ percentiles

of $O_3$, NO, $NO_2$, wind speed, temperature, and relative humidity for the entire campaign, and $J(O^1D)$, $J(HCHO\_M)$, and

$J(HCHO\_R)$ from sunrise to sunset during the campaign.

   $J(O^1D)$ showed a normal diurnal variation, the daytime $J(O^1D)$ from sunrise to sunset is characterized with an average of

$1.23 \times 10^{-5}$ $s^{-1}$ and a $10^{th}$ and $90^{th}$ percentile of $1.24 \times 10^{-6}$ $s^{-1}$ and $3.08 \times 10^{-5}$ $s^{-1}$, respectively. Ambient temperature was

characterized with an average of 26 ℃ and a $10^{th}$-$90^{th}$ percentile range of 23-30 ℃, and RH showed an average of 83% and a

$10^{th}$-$90^{th}$ percentile range of 62-98%, which is consistent with the typical features of the Plum Rain Season. The prevailing

winds were from southeast, with an average speed of 1.8 m $s^{-1}$.

   The concentration of $O_3$ was characterized with an average of 31 ppbv and varied in a $10^{th}$-$90^{th}$ percentile range of 8-59

ppbv. The $10^{th}$-$90^{th}$ percentile concentrations of NO and $NO_2$ were 2-8 and 7-23 ppbv, with average values of 6 and 14 ppbv,

respectively. The daily maximum 1 h-$O_3$ concentration of five days during the campaign have exceeded Class I of China

National Air Quality Standards (CNAAQS) i.e., an hourly average of 160 μg $m^{-3}$ (~75 ppbv). Compared with a previous

campaign operated at the DSL site in the summer of 2020, where the average concentrations of $O_3$, NO, and $NO_2$ were 37

ppbv, 4 ppbv, and 18 ppbv, respectively (Yang et al., 2022), the concentrations of these pollutants in this observation were

similar.

   HCHO mixing ratios ranged up to the maximum of 9.4 ppbv, with the average value of 2.2±1.8 ppbv (one standard deviation)

and the median value of 1.8 ppbv. This value was comparable to that (2.2 ppbv) reported in the urban area of New York (USA)

in summertime (Lin et al., 2012), but lower than those (5.07 and 5.0 ppbv, respectively) observed in the urban area of Shanghai

(China) and Shenzhen (China) (Ho et al., 2015; Wang et al., 2017). However, the HCHO concentration level at the DSL site

was higher than those (1.5, 1.34, 1.1, 1.1, and 0.4 ppbv, respectively) reported in remote areas in Mazhuang Town (China),

Whiteface Mountain (USA), Ineia (Cyprus), Hohenpeißenberg (Germany) and Hyytiälä (Finland) (Xiaoyan et al., 2010; Zhou

et al., 2007; Nussbaumer et al., 2021).

In Figure 2, $O_3$ and $O_x$ showed relatively high abundances while the concentration of traffic-related species like $NO_x$ was

low. This hints that HCHO at the observation site was potentially associated more with the secondary sources. Figure 3a further

illustrates the diurnal variations of HCHO, $O_x$ and $NO_x$ at the DSL site during the campaign. Both HCHO and $O_x$ exhibited

strong diurnal variations during the field measurements, which reached the peak at the early afternoon, decreased gradually in

the afternoon, and remained flat at night. The good correlation between hourly concentrations of HCHO and $O_x$ (R=0.68), as

shown in Figure 3b, indicates important contributions of secondary sources to HCHO. By contrast, the variations of HCHO

and $NO_x$ in Figure 3a are less parallel. $NO_x$ reached its maximum in the morning rush hours at about 5:00 LT (local time), then

decreased till noon, and gradually increased to relatively high values after 19:00 LT, which likely coincided with the traffic



volumes on the highroad. The correlation (R=0.02) between HCHO and $NO_x$ was really poor as shown in Figure 3c. These observations indicate that HCHO at the DSL site was dominantly influenced by secondary rather than primary sources.

We examined the ratios of HCHO and $NO_x$, as shown in Eq. (1) (Lin et al., 2012) to quantify the secondary HCHO. The primary ratio between the initial mixing ratios of HCHO and $NO_x$ in the fresh emissions $[HCHO/NO_x]_{pri}$ was assumed to be the 10th percentile value (0.02) of all the measured values of $HCHO/NO_x$ during our campaign (Wang et al., 2020),

$$f_{HCHO} = \left(\left[\frac{HCHO}{NO_x}\right]_a - \left[\frac{HCHO}{NO_x}\right]_{pri}\right)\bigg/\left[\frac{HCHO}{NO_x}\right]_a \tag{1}$$

where $f_{HCHO}$ represents the fraction of secondary HCHO, $\left[\frac{HCHO}{NO_x}\right]_a$ and $\left[\frac{HCHO}{NO_x}\right]_{pri}$ represents the HCHO-to-$NO_x$ ratios in the

ambient air and fresh emissions, respectively.

Figure 4 reveals our estimation of average contributions of secondary HCHO during a day. Secondary HCHO exceeded primary HCHO (including background HCHO) nearly all the time, except for 5:00-6:00 LT, when there were more traffics on the highroad. Secondary HCHO dominated over the DSL site from 10:00 to 18:00 LT, with an obvious enhancement at noon and in the afternoon, reaching an average daily maximum of 78% at 17:00 LT, then gradually decreased, and remained flat at

about 55% during the nighttime. On average, daily secondary HCHO was estimated to contribute to approximately 70% of the total ambient HCHO.

**3.2 HCHO production from VOC oxidation**

HCHO is secondarily produced through oxidation of a wide range of atmospheric VOCs by oxidants including OH and $O_3$. Hence, with HCHO yields reported in literatures for these oxidation processes and the concentrations of the parent VOCs and

oxidants, the chemical production rate of HCHO can be estimated as shown in Eq. (2) (Lee et al., 1998; Sumner et al., 2001; Choi et al., 2010; Lin et al., 2012).

$$P(HCHO) = \sum_i \sum_j \left(\gamma_{ij} k_{ij} [VOC]_J [Oxindant]_i\right) \tag{2}$$

where $i$ denotes the $i$th kind of oxidant such as OH or $O_3$, $j$ denotes the $j$th VOC species that produces HCHO through its oxidation, and $k_{ij}$ and $\gamma_{ij}$ represent the reaction rate coefficient and the corresponding HCHO yield for the reaction between the

$i$th oxidant and the $j$th VOC, respectively.

Over 60 VOC species were detected during the campaign, but only 24 VOC species whose oxidation will produce HCHO were used in the calculation of HCHO production rates, including 12 alkanes, 1 aromatic, 9 alkenes and 2 OVOCs. Concentrations of theses 24 VOCs are summarized in Table S3, and the corresponding reaction rate coefficients and HCHO yields taken in the calculation are described in Table S4. The VOC species considered are quite comprehensive, though

methanol, acetone and acetaldehyde are not available in this study. The key processes to form HCHO from acetone and acetaldehyde are quite similar with that from methane, with the same intermediate product of methyl peroxy radicals ($CH_3O_2$),





but they contribute much less to HCHO production in contrast to methane according to a previous study (Nussbaumer et al., 2021).The contribution of methanol was less than half of methane in a previous study (Nussbaumer et al., 2021). Since there are no obvious direct emissions of these VOCs near the site, we consider their absence would not influence our results

considerably.

There was not a direct measurement of OH radical concentrations during this campaign, and thus we adopted an empirical equation that has been suggested for OH concentration estimations in four Chinese megacities including Shanghai, as shown in Eq. (3) (Liu et al., 2020; Tan et al., 2019a; Fan et al., 2021),

$$[OH] = J(O^1D) \times 3 \times 10^{11}\ molecules\ cm^{-3} \tag{3}$$

where $J(O^1D)$ denotes photolysis frequencies of O[1]D. We compared our calculation results (Table S5) with those from another recommendation (Sect. S1) (Ehhalt and Rohrer, 2000), and their good correlation (R=0.97) and a slope close to 1 in Figure S2 validates our estimates. The uncertainties of OH concentration from the calculation were estimated to be 20%, and this uncertainty was used to estimate those in the production rates and loss rates of HCHO (Tan et al., 2019a; Rohrer and Berresheim, 2006). Figure 5 presents the profile of the calculated HCHO production rates during the whole campaign with a

time resolution of 1 hr. Overall, alkenes oxidation by OH radicals contributed the most to secondary HCHO production, accounting for 66.3%, followed by OH-radical initiated reactions with alkanes and aromatics (19.0%) and reactions of OVOCs (8.7%), while ozonolysis of alkenes contributed by 6.0%, which was the smallest contribution reaction pathway. The average of calculated secondary HCHO production rate is 0.73 ppbv h$^{-1}$, with a 90$^{th}$ percentile of 2.42 ppbv h$^{-1}$ and a 10$^{th}$ percentile of 0.01 ppbv h$^{-1}$. Peaks of secondary HCHO production rates were usually observed at noon, and the rates showed obvious diurnal

cycles. On the other hand, the rates varied significantly during the campaign because secondary HCHO production relies heavily on the weather condition i.e., photochemical reactions are usually much more active in the sunny days than in the cloudy and rainy period. As we have mentioned before, there were obvious weather variations during the campaign. Therefore, we divided our campaign into the sunny period (including 12 days) and the cloudy and rainy period (13 days) for further investigation. Comparison of the secondary HCHO production between the sunny period and the cloudy and rainy period is

shown in Table 1.

In Figure 6, relative contributions to HCHO production from various processes during the sunny and the cloudy and rainy periods, respectively, are shown, together with the top 10 VOC species that contributed the most in each period, which in total yielded more than 90% of the overall HCHO. During the sunny days (Figure 6a), HCHO production was dominated by the reactions of alkenes and OH radicals, accounting for 64.8%, followed by OH radical-initiated reactions with alkanes and

aromatics (19.5%), OH radical-initiated reactions with OVOCs (10.4%), and ozonolysis of alkenes (5.3%). As the graph shows, 32.3% of the secondary HCHO production came from isoprene oxidation (by both OH radical and O$_3$), where OH oxidation

of isoprene (30.6%) overwhelmed. The other main contributors in the sunny days were associated with OH radical-initiated reactions with ethene (19.4%), methane (12.9%), propene (10.6%), and MVK (8.0%), which together with isoprene represented more than 80% of the overall HCHO production.

For the cloudy and rainy period (Figure 6b), the relative contribution to secondary HCHO from OH radical-initiated reactions with alkenes increased a little, accounting for 68.3%, while OVOCs oxidation by OH radicals decreased to 5.8%. MVK and MACR are known as the major intermediate products generated from isoprene oxidation. The less intensive solar radiation in the cloudy and rainy days influences both the abundance and the oxidation processes of MVK and MACR to form HCHO, leading to their declined fraction to HCHO production (Gong et al., 2018; Guo et al., 2012; Gu et al., 2022). Meanwhile,

the total contribution from isoprene oxidation decreased to 21.0% due to the combination of lower isoprene concentrations and OH abundances. In the cloudy and rainy days, the dominant pathways to HCHO production were OH radical-initiated reactions with ethene, isoprene, propene and methane, yielding 30.2%, 21.0%, 13.2% and 11.9%, respectively, of the total HCHO production from VOCs we have measured.

The average diurnal patterns of secondary HCHO production showed clear differences between the sunny and the cloudy

and rainy periods, as shown in Figure 7. During the sunny period (Figure 7a), HCHO production rates displayed a strong diurnal cycle, with a peak of 3.80 ppbv h$^{-1}$ observed at 13:00 LT when photochemical reactions were intense, and were roughly constant at about 0.03 ppbv h$^{-1}$ during nighttime (from 19:00 to 5:00 LT next day). During this low-rate period, about 98% of the HCHO production came from ozonolysis of alkenes, since the estimated average nighttime concentration of OH radicals was lower than 1500 molecules cm$^{-3}$, i.e., $5.58\times10^{-5}$ pptv, while that of O$_3$ was still as high as $5.68\times10^{11}$ molecules cm$^{-3}$, i.e.,

21.15 ppbv. After sunrise at 5:00 LT, HCHO production rates increased dramatically, reaching the maximum of 3.80 ppbv h$^{-1}$ at 13:00 LT, and then reduced until sunset at around 18:00 LT. Although O$_3$ concentration was higher than that of OH radicals during the daytime, the rate constants for reactions of alkenes with O$_3$ are several orders of magnitude lower than those with OH, resulting in the dominant HCHO formation by alkenes oxidation with OH.

In the cloudy and rainy period (Figure 7b), nighttime HCHO production rates were almost equivalent to those in the sunny

days, and also dominantly came from alkene ozonolysis. Secondary HCHO production rates began to rise after 5:00 LT, peaked at 10:00 LT (1.82 ppbv h$^{-1}$), maintained high (~1.69 ppbv h$^{-1}$) until 13:00 LT, and then started to fall to low values at night. This trend is consistent with the variation of the photolysis frequencies, which remained the highest values between 10:00-13:00 LT in the rainy and cloudy days, whereas they kept growing after sunrise and peaked at 13:00 LT in the sunny days. The diurnal average HCHO production rates in the cloudy and rainy days (0.51 ppbv h$^{-1}$) were nearly half of the average in

the sunny days (0.97 ppbv h$^{-1}$).



By applying the root square propagation of uncertainties in the reaction rate coefficients and corresponding HCHO yields for reactions between VOCs and oxidants, measurements of 24 VOCs and ozone, and estimations of OH (Table S6), the total uncertainties of the HCHO production rates were estimated to be 25.9% in the sunny period, and 21.0% in the cloudy and rainy period.

### 3.3 HCHO sinks

Reactions (R1) ~ (R3) show the dominant daytime chemical loss processes of HCHO, i.e, direct oxidation by OH radicals, and two different photolysis pathways. The overall HCHO loss rate by photolysis can be calculated from measured HCHO concentration and its photolysis rate constants, J(HCHO_M) and J(HCHO_R) (shown in Table S1). These major daytime sinks of HCHO ultimately produce hydroperoxy ($HO_2$) radicals. As reported in previous studies, HCHO represents an important source of $HO_2$ radicals in the atmosphere (Mahajan et al., 2010; Tan et al., 2019a, b).

$$HCHO + OH + O_2 \rightarrow CO + HO_2 + H_2O \qquad \text{(R1)}$$

$$HCHO + h\nu \rightarrow CO + H_2 \qquad \text{(R2)}$$

$$HCHO + h\nu + 2O_2 \rightarrow CO + 2HO_2 \qquad \text{(R3)}$$

Compared with daytime, photolysis frequencies and OH radical concentrations at night are really low so that oxidation of HCHO by OH radicals and photolysis are ineffective sinks. Instead, HCHO dry deposition become the most important nocturnal removal process at night, which depends on both its loss at a surface (described by a surface resistance) and transport to the surface (Fischer et al., 2019; Nussbaumer et al., 2021; Choi et al., 2010; Nguyen et al., 2015; Sumner et al., 2001; Anderson et al., 2017). The HCHO deposition velocity $v_d$ can be estimated from its nighttime concentration decrease (Nussbaumer et al., 2021; Fischer et al., 2019). An average loss rate constant $k_d$ was determined from the HCHO concentration decline from 21:00-01:00 LT divided by the average HCHO concentration during this time interval according to Eq. (4).

$$k_d(HCHO) = \frac{\frac{d[HCHO]}{dt}}{[HCHO]_{av}} \qquad \text{(4)}$$

Then the HCHO deposition velocity could be calculated by Eq. (5).

$$v_d(HCHO) = \frac{k_d \times BLH}{x} \qquad \text{(5)}$$

where BLH denotes the boundary layer height. To consider the inconsistent mixing of the boundary layer at night, the factor $x$ is equal to 2, assuming a linear increase in the HCHO mixing ratio with height in the nocturnal boundary layer (Shepson et al., 1992). During the day, $x$ is set to be 1 for a boundary layer that is well mixed (Fischer et al., 2019; Nussbaumer et al., 2021). Note that this estimation of the dry deposition loss is a lower limit, since it neglects thermally driven turbulence and deposition caused by stomatal uptake by vegetation (Fischer et al., 2019; Nguyen et al., 2015). Figure S3 shows an example of one of the evenings during which HCHO decay followed an apparent first-order kinetics. We have performed this calculation





for 9 nights and an overview of the 9 nights can be found in Figure S4. We finally estimated $v_d$ (night) = 0.52 cm/s and $v_d$ (day)

= 1.04 cm/s. Table 2 compares dry deposition rates of HCHO reported in previous studies to our estimates, which turn out to

be quite similar (DiGangi et al., 2011; Ayers et al., 1997; Stickler et al., 2007; Nussbaumer et al., 2021; Sumner et al., 2001;

Choi et al., 2010).

By our estimation, the role of $NO_3$ radicals in HCHO removal at the DSL site was negligible (Sect. S2). Therefore,

calculation of the HCHO loss could be expressed as Eq. (6).

$$L(HCHO) = L_{HCHO+OH} + L_{HCHO+h\nu} + L_{deposition}$$

$$= [HCHO] \times ([OH] \times k_{HCHO+OH} + (J(HCHO\_M) + J(HCHO\_R)) + \frac{v_d(HCHO)}{BLH}) \qquad (6)$$

The profile of the calculated HCHO loss rates during the campaign is shown in Figure S5, with an average loss rate of 0.49

ppbv h$^{-1}$. Comparison of HCHO loss rates between the sunny period and the cloudy and rainy period is shown in Table 1.

In Figure 8, the diurnal average HCHO loss rates in both the sunny period and the cloudy and rainy period showed significant

diurnal cycles. Daily average loss rates of dry deposition did not show obvious diurnal cycle, which remained relatively

constant ranging from 0.08 ppbv h$^{-1}$ to 0.40 ppbv h$^{-1}$ in the sunny period and 0.04 ppbv h$^{-1}$ to 0.12 ppbv h$^{-1}$ in the cloudy and

rainy period, owning to co-variation of the concertation of HCHO and BLH. During nighttime in both periods, HCHO loss

rates were dominated by dry deposition, when the contributions of photolysis and reactions with OH were so small that they

could be neglected. After sunrise, loss rates of photolysis and the reaction with OH radicals began to rise, and reached the

maximum at noon. The diurnal maximum loss rates in the sunny period was 2.75 ppbv h$^{-1}$, about 3 times larger than that in the

cloudy and rainy period (0.66 ppbv h$^{-1}$), both of which occurred at 13:00 LT. After the peak, loss rates of photolysis and the

reaction with OH in both phases continued to fall and remained low at night. The diurnal average loss rates of HCHO were

0.78 ppbv h$^{-1}$ and 0.22 ppbv h$^{-1}$ for the sunny period and the cloudy and rainy period, respectively. Dry deposition played a

more important role in HCHO loss in the cloudy and rainy period, accounting for 34.1% of the loss of HCHO, whereas

photolysis contributed by 32.8% and the reaction with OH radicals contributed by 33.1%. Reaction with OH radicals was the

dominant contributor to HCHO loss in the sunny period, which represented 42.2% of the total HCHO loss, followed by

photolysis (39.2%), and dry deposition (18.6%).

The total uncertainties of HCHO loss rates resulted from the uncertainties in the reaction rate coefficients for VOCs and OH

radicals, measurements of HCHO, photolysis frequencies, and PBL, and estimations of OH (Table S7), were 28.9% in both

the sunny period and the cloudy and rainy period.


### 3.4 HCHO net production

We investigated the daily profiles of hourly averages of HCHO production and loss rates throughout the campaign. Net production of HCHO can be calculated as Eq. (7).

$$Net(HCHO) = P(HCHO) - L(HCHO) \qquad (7)$$

Figure 9a shows that our net HCHO production are in good agreements with the observed HCHO rate of change ($\frac{d[HCHO]}{dt}$) throughout the day in the sunny period, indicating that HCHO was dominantly secondary produced at the DSL site, which can be approximated by oxidation of the 24 hydrocarbons we considered in calculation. Thus, transport processes and primary emissions can likely be excluded. During nighttime, the measured rate of HCHO change oscillated around zero, when the calculated production almost completely balanced the loss term. After 7:00 LT, HCHO production exceeded its loss, leading to positive net HCHO production values, and were in line with the increasing trend of the HCHO concentration. After 15:00 LT, HCHO loss began to transcend its production, resulting in the decline of HCHO abundance. Total uncertainties in HCHO production rates (25.8%) and loss rates (28.9%) also led to overlapped uncertainty ranges for the HCHO production and loss, as shown in Figure S6a. Thus, the calculated production could be very close to the loss, which is consistent with the observed rate of HCHO change.

On the other hand, in the cloudy and rainy days, the observed HCHO concentration remained relatively steady while calculated HCHO production prevailed over its loss, leading to a net production of about 1 ppbv h$^{-1}$ during 8:00-13:00 LT, as shown in Figure 9b. The difference between the net production and the observed HCHO rate of change suggests either a missing loss term or an overestimated production. Since significant differences were still observed from 8:00-13:00 LT in Figure S6b even when uncertainties in the HCHO production (21.0%) and loss rates (28.9%) were taken into account, a real missing loss term was more likely there. There was significantly more rainfall at daytime in the cloudy and rainy days than in the sunny days during our campaign, and thus we consider the missing loss process might be wet deposition, which has been reported as a dominant removal of the total deposition (i.e. dry deposition and wet deposition) during the rainy season (Seyfioglu et al., 2006). Indeed, HCHO is readily soluble in cloud and rainwater with its high Henry's law constant ($\sim5.5\times10^3$ M atm$^{-1}$) and could be efficiently converted to formic acid in warm cloud droplets (Allou et al., 2011; Chebbi & Carlier, 1996; Franco et al., 2021). Unfortunately, we did not collect rain samples during our campaign so that we do not have an access to further evaluate this assumption.

### 4 Conclusions

In this study, ambient HCHO measurements, together with a number of VOC species, were conducted from June 10 to July 4 in 2021 at the DSL site in Shanghai. During the campaign, the average HCHO concentration was 2.2 ± 1.8 ppbv. The good



correlation of HCHO and $O_x$ (R=0.68) and the values of HCHO/$NO_x$ indicates that secondary sources played an important role in the local HCHO formation. 24 VOC species were considered in the calculation of secondary HCHO production, which shows that the dominant HCHO precursors were isoprene, ethene, methane and propene. In the sunny period, isoprene oxidation by OH radicals contributed the most, whereas reactions of ethene with OH radicals became the most important path

to the HCHO production in the cloudy and rainy period. The diurnal average secondary HCHO production rates were 0.97 and 0.51 ppbv h$^{-1}$ for the sunny period and the cloudy and rainy period, respectively. For the HCHO loss estimation, HCHO photolysis, reactions with OH radicals, and dry deposition, were considered, where loss rates due to photolysis and reactions of OH radicals were significantly larger than that of dry deposition in the sunny period, but these three terms were nearly equivalent in the cloudy and rainy period. The diurnal average loss rates of HCHO were 0.78 ppbv h$^{-1}$ and 0.22 ppbv h$^{-1}$ for

the sunny period and the cloudy and rainy period, respectively. Net HCHO production were in good agreements with the observed HCHO rate of change throughout the sunny days, indicating that HCHO was approximately produced by oxidation of the 24 VOC species we considered at the DSL site during the campaign.

In summary, our results reveal the important role of secondary formation of HCHO at the suburb of Shanghai, where alkenes are likely the key precursors for HCHO. We provide a HCHO budget based on a comprehensive observation of HCHO

precursors, which has rarely been conducted in previous studies. Meanwhile, we found evidences for missing loss processes of HCHO in the cloudy and rainy days, which might be attributed to the HCHO wet deposition, and this may be an important loss term in rainy days and should be further investigated.

*Data availability.* The data used to support the conclusions in this study are available at a public data repository of Figshare

via https://doi.org/10.6084/m9.figshare.20218133.v3

*Author contributions.* LW designed the study. YZW, JH, GY, YWW, SW, and QF conducted the field campaign. YZW, GY, YWW, and LHW carried out laboratory experiments. JH, SW, QF, and YL provided technical support. YZW analysed the data. YZW and LW wrote the paper with contributions from all of the other co-authors.


*Competing interest.* The authors declare that they have no conflicts of interest.

*Acknowledgements.* This research was supported by the National Natural Science Foundation of China (21925601, 92044301, 92143301 and 22127811), and Shanghai Municipal Bureau of Ecology and Environment (2021-29).






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



**Table 1. Comparison of the HCHO concentration, production rates, and loss rates between the sunny period and the cloudy and rainy period.**

| Period | Total | Sunny phase | Cloudy and rainy phase |
|---|---|---|---|
| Date | 6.10 ~ 7.4 | 6.11, 6.12, 6.15, 6.16, 6.21 ~ 6.25, 6.29 ~ 7.1 | 6.10, 6.13, 6.14, 6.17 ~ 6.20, 6.26 ~ 6.28, 7.2 ~ 7.4 |
| Average conc. (ppbv) | 2.2 | 2.8 | 1.7 |
| $10^{th}$ percentile of conc. (ppbv) | 0.4 | 0.3 | 0.5 |
| $90^{th}$ percentile of conc. (ppbv) | 4.9 | 6.2 | 3.4 |
| Average production (ppbv h$^{-1}$) | 0.73 | 0.97 | 0.51 |
| $10^{th}$ percentile of production (ppbv h$^{-1}$) | 0.01 | 0.01 | 0.01 |
| $90^{th}$ percentile of production (ppbv h$^{-1}$) | 2.42 | 3.15 | 1.53 |
| Average loss (ppbv h$^{-1}$) | 0.49 | 0.78 | 0.22 |
| $10^{th}$ percentile of loss (ppbv h$^{-1}$) | 0.02 | 0.01 | 0.02 |
| $90^{th}$ percentile of loss (ppbv h$^{-1}$) | 1.46 | 2.64 | 0.5 |



**Table 2. A summary of HCHO deposition velocities.**

| Site | Time | Daytime dry deposition velocities (cm/s) | Nighttime dry deposition velocities (cm/s) | Reference |
|---|---|---|---|---|
| Shanghai, China Suburban area | Summer 2021 | 1.04 | 0.52 | This study |
| Hohenpeißenberg, Germany Suburban area | Summer 2012 | 0.94 | 0.47 | (Nussbaumer et al., 2021) |
| Colorado, America Suburban area | Summer 2010 | 0.39 | 0.18 | (DiGangi et al., 2011) |
| California, America Suburban area | Autumn 2007 | 1.5 | 0.84 | (Choi et al., 2010) |
| Michigan, America Suburban area | Summer 1998 | 1.5 | 0.65 | (Sumner et al., 2001) |
| Cape Grim, Australia Suburban area | Winter 1993 | -- | 0.5 | (Ayers et al., 1997) |



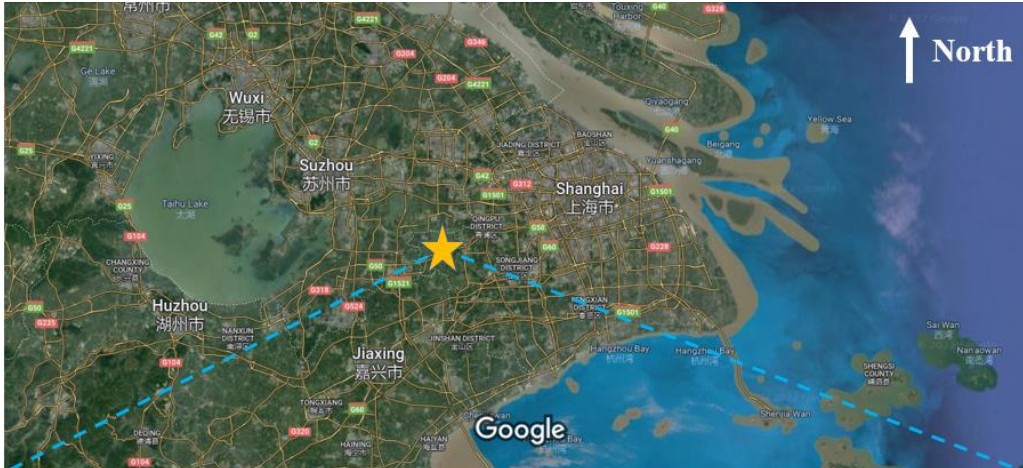

Imagery ©2022 TerraMetrics, Map data ©2022    10 mi

Imagery ©2022 CNES / Airbus, Maxar Technologies, Map data ©2022    500 ft

**Figure 1. A Topography map (from ©Google Maps) of the region around the Dianshan Lake (DSL) Air Quality Monitoring Supersite (31.10°N, 120.98°E).**






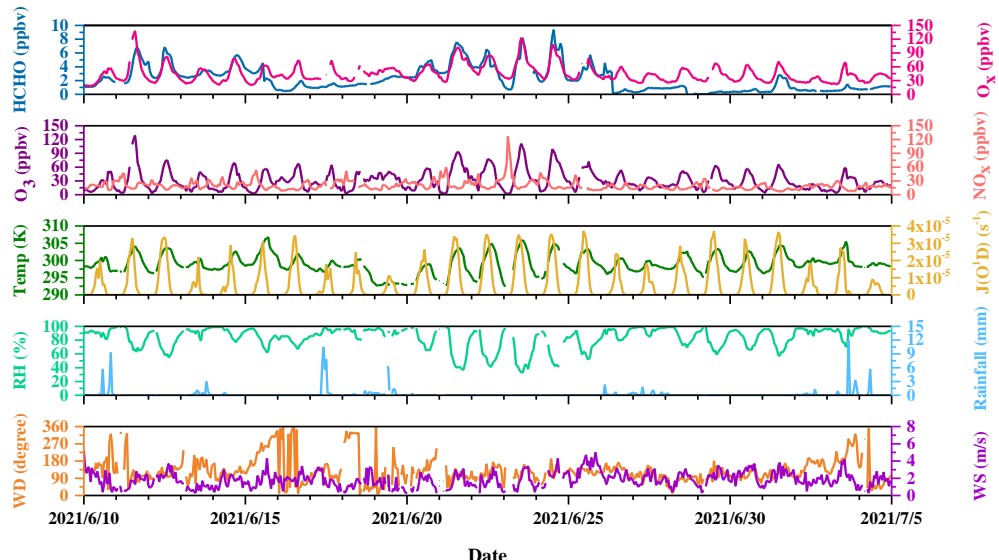

**Figure 2.** Time profiles of HCHO, $O_x$ (=$NO_2$+$O_3$), $O_3$, $NO_x$, photolysis frequencies of $O^1D$ (J($O^1D$)) and meteorological parameters during the campaign.



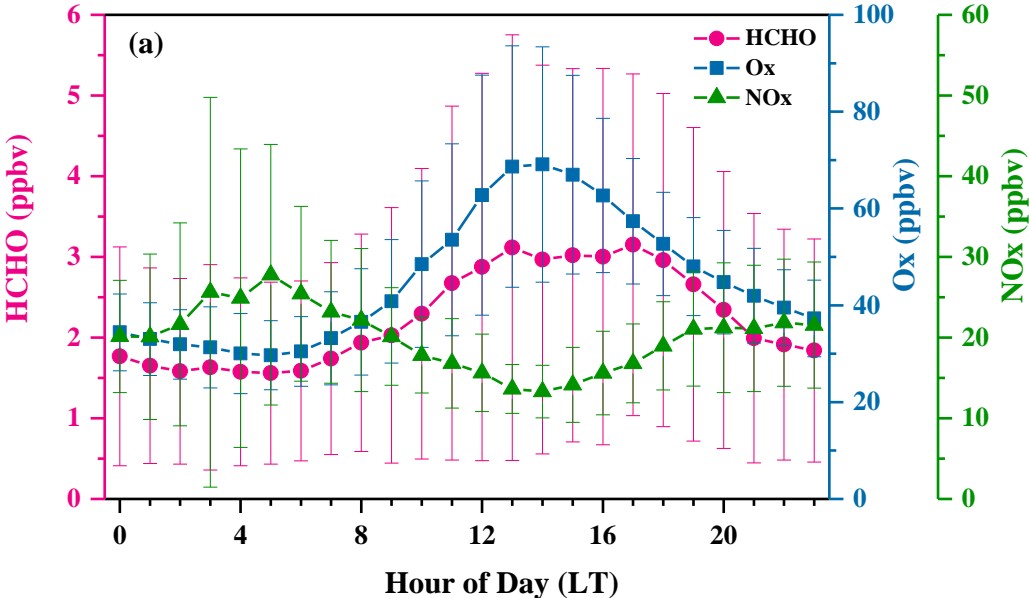

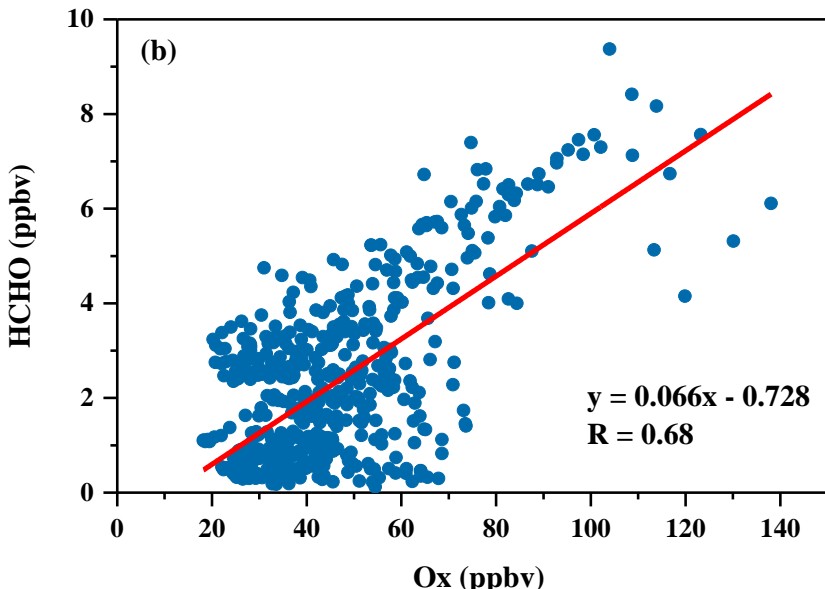





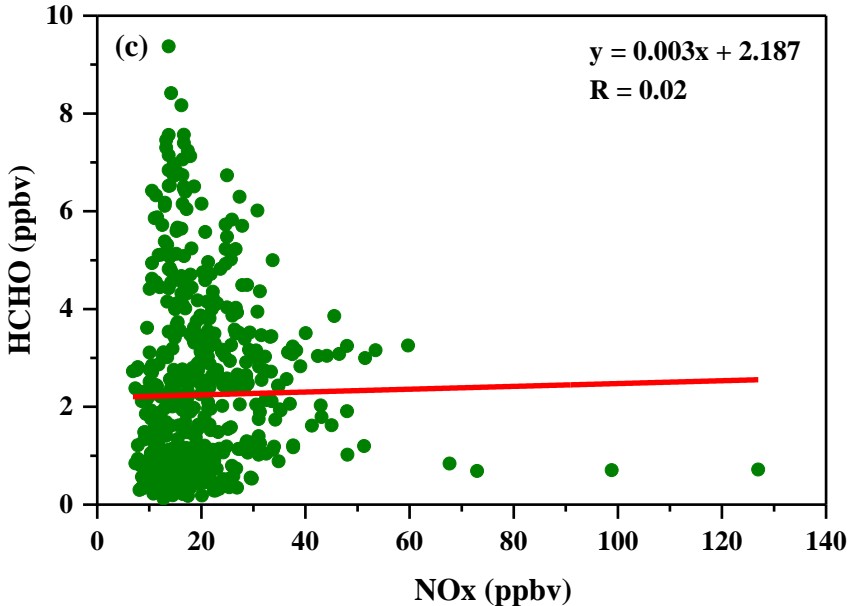

**Figure 3. (a) Diurnal patterns of HCHO, $O_x$, and $NO_x$, with the error bar indicating one standard deviation; (b) Correlation between the measured concentrations of HCHO and $O_x$; (c) Correlation between the measured concentrations of HCHO and $O_x$.**





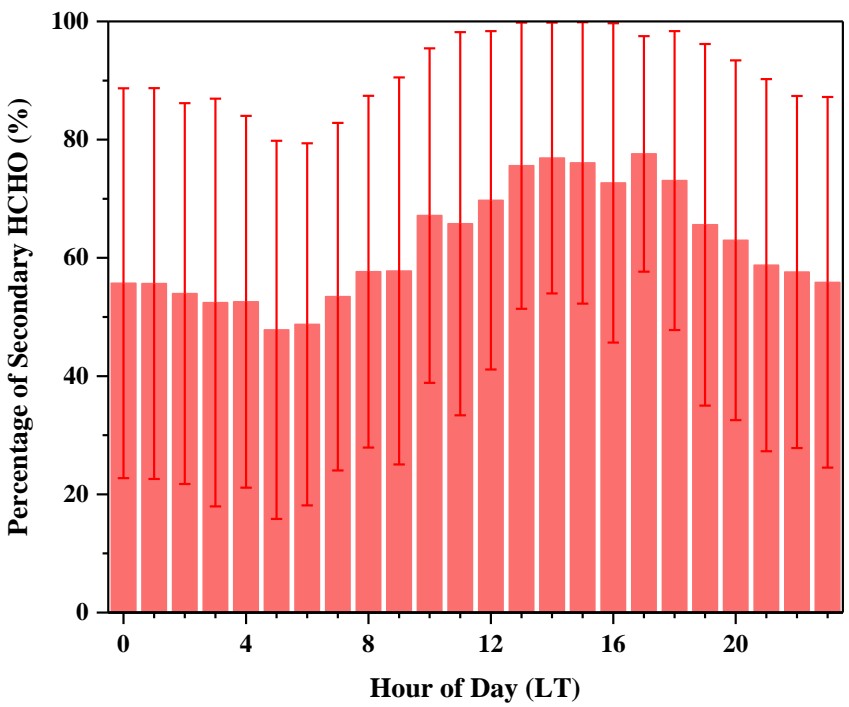

**Figure 4. Diurnal contribution of secondary HCHO, with the error bar indicating one standard deviation.**






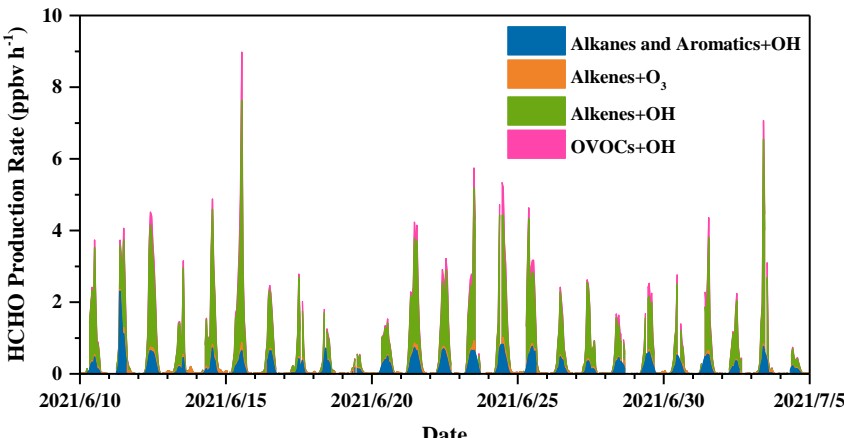

**Figure 5. Time profiles of the calculated HCHO production rates during the campaign.**





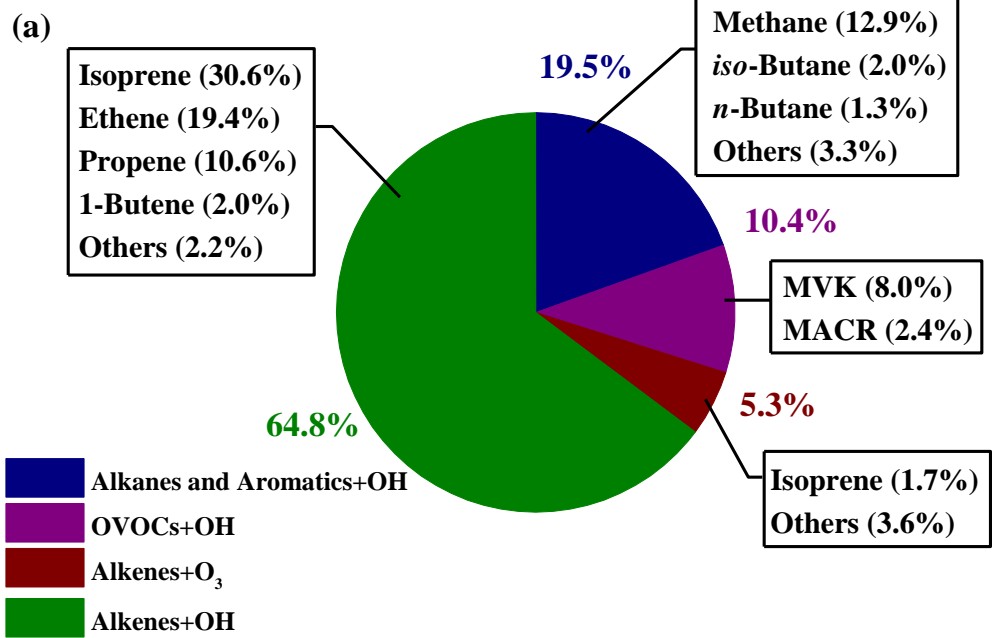

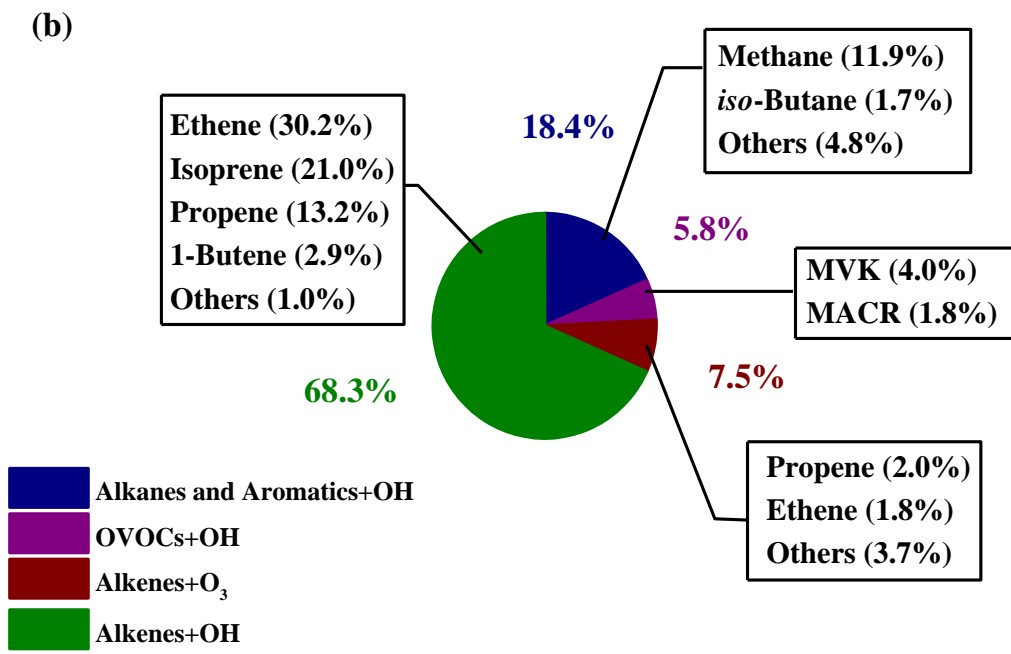

**Figure 6**. Relative contributions to HCHO production rates in (a) the sunny period and (b) the cloudy and rainy period. The relative contributions to HCHO production are shown for the top 10 VOC species in each phase. The area of pie charts is in proportion to the calculated HCHO production rates in two periods.


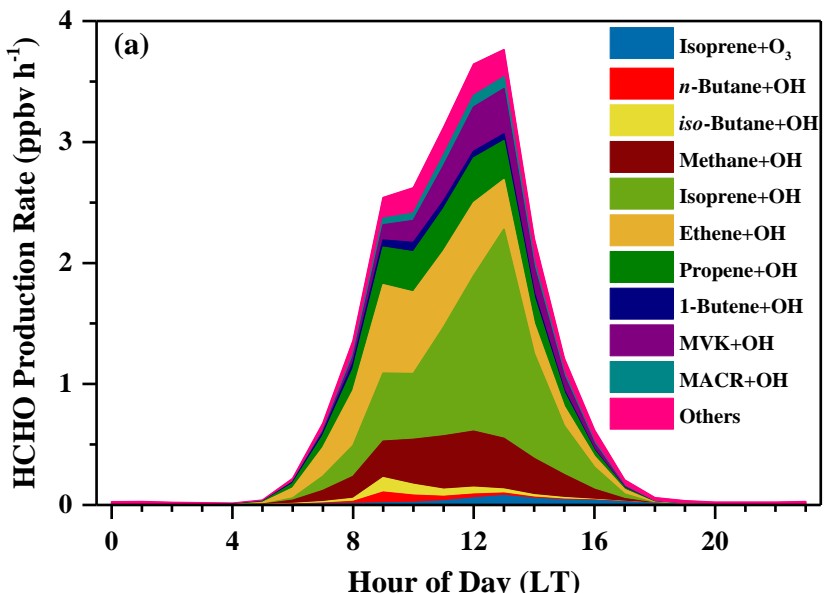

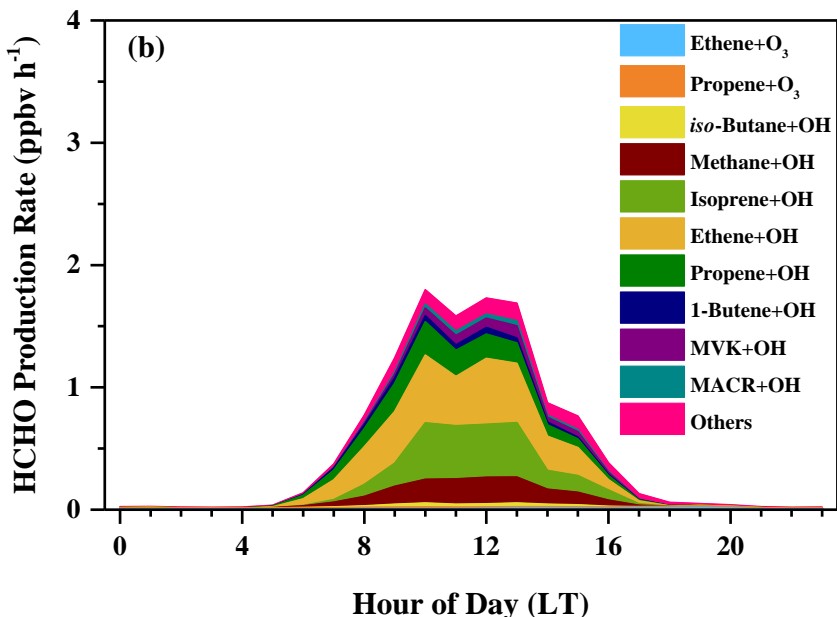

Figure 7. Average diurnal variations of HCHO production rates (ppbv h$^{-1}$) from the OH-initiated and O$_3$-initiated oxidation during (a) the sunny period and (b) the cloudy and rainy period.





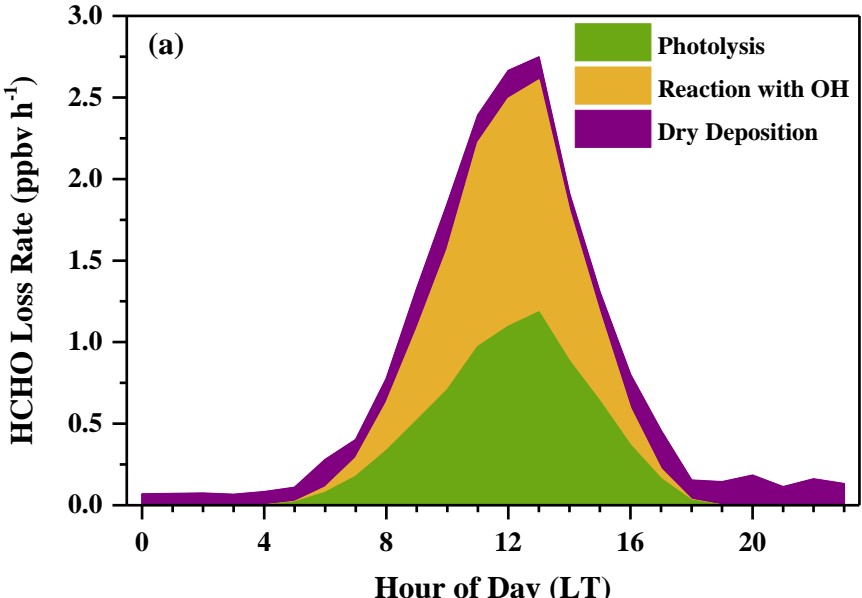

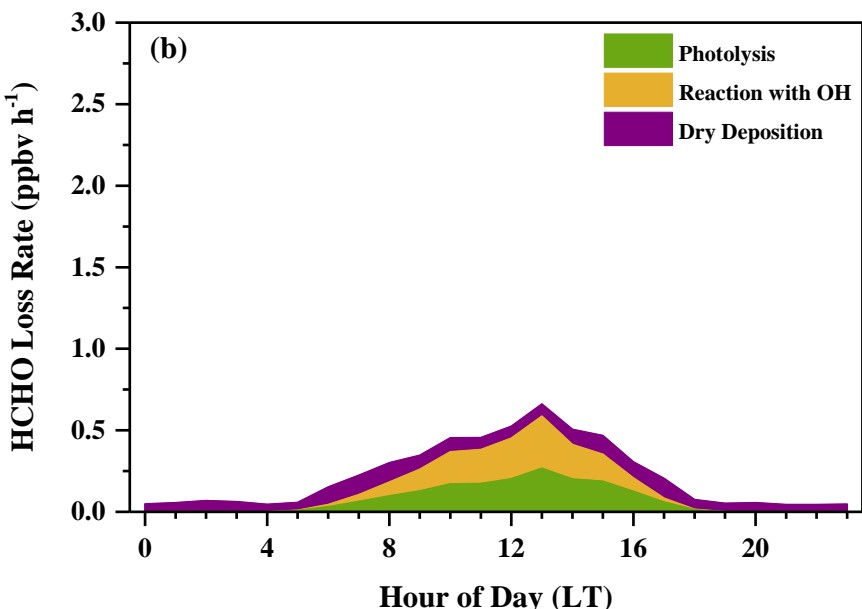

**Figure 8**. Average diurnal variations of HCHO loss rates (ppbv h⁻¹) from photolysis, reaction with OH radicals and dry deposition during (a) the sunny period and (b) the cloudy and rainy period.





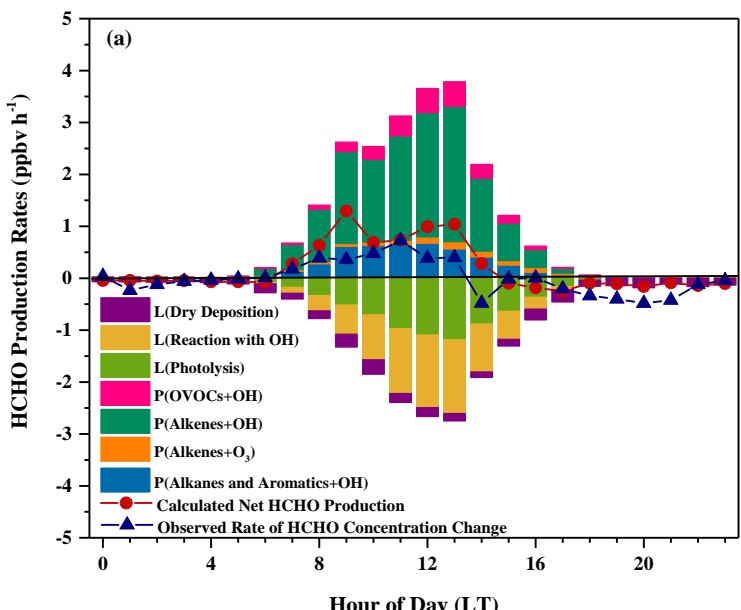

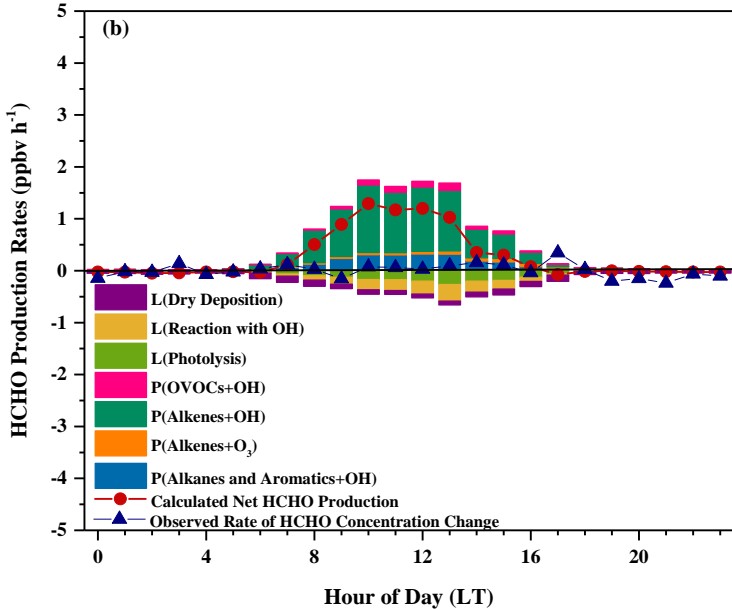

**Figure 9. Total average daytime production rates, loss rates, the calculated net production, and the observed rate of HCHO concentration change of HCHO for (a) the sunny and (b) the cloudy and rainy period.**
