# Peer review of "Measurement report: Production and loss of atmospheric formaldehyde at a suburban site of Shanghai in summertime"

_Atmospheric Chemistry and Physics, 2022_

## Author Comment (AC1)

**A point-to-point response to reviewers' comments**

We are very grateful to the helpful and insightful comments from the reviewers, and have carefully revised our manuscript accordingly. In the following point-to-point response, reviewers' comments are repeated in *italics* whereas our responses are in plain texts labelled with **[R]**. Line numbers in the responses correspond to those in the revised manuscript (the version with all changes accepted). Modifications to the manuscript are in blue.

**Reviewer #1** (Formal Review for Author):

*The manuscript entitled "Measurement report: Production and loss of atmospheric formaldehyde at a suburban site of Shanghai in summertime" by Wu Yizhen et al. reports the observation-based budget analysis of formaldehyde in a suburban area of Shanghai. Although the authors showed convincible measurements of HCHO as well as VOCs, NOx, O3, etc., the estimation of HCHO sources and sinks might suffer from major flaws (as I will explain in the following). Moreover, the concentration level and the diurnal variation of HCHO at the specific site are not beyond our common knowledge. I don't see the add-value of the reported data. Therefore, I would recommend the publication only if the authors well addressed my following comments.*

**R:**

We appreciate the comments from Reviewer #1 and have revised our manuscript accordingly as shown below.

*(1) The estimation of secondary HCHO looks quite unreliable. In the introduction part, the authors listed several previous studies using different methods to discriminate primary and secondary HCHO. However, in the manuscript, HCHO/NOx ratio is used. It is a simple one compared to PCA, multi-linear regression, etc.. But it may have large uncertainties. Especially, compared to the one proposed by de Gouw et al. (2005), HCHO/NOx dose not well consider photochemical aging processes. For example, HCHO and NO could be high in fresh emissions (e.g., vehicle exhaust) while HCHO and NO2 could be high in an aged plume, this would result in same HCHO/NOx ratio for both cases. Therefore, the rationale / validity of applying HCHO/NOx method for the observation site is necessary, and yet is missing in the manuscript. The same for the uncertainty analysis. Moreover, since the authors have a comprehensive dataset of various VOCs, OVOCs, etc., it is possible to apply de Gouw's method. It would be better to see the comparison of the secondary contribution derived from the two methods.*

**R1:**

NO$_x$ and C$_2$H$_2$ are usually regarded as primary pollutants mostly from automobile exhaust (de Gouw et al., 2005; Lin et al., 2012; Zhang et al., 2013; Dutta et al., 2010). Since secondary HCHO is not totally independent from NO$_x$ (Wolfe et al., 2019) as Referee #2 pointed out, we now use C$_2$H$_2$, which is a relatively inert VOC ($k_{C2H2-OH}=5\times10^{-30}$ cm$^3$ molecule$^{-1}$ s$^{-1}$ and $k_{C2H2-O3}=1\times10^{-20}$ cm$^3$ molecule$^{-1}$ s$^{-1}$ (IUPAC Task Group on Atmospheric Chemical Kinetic Data Evaluation, available at https://iupac-aeris.ipsl.fr/index.html#)) and comes mostly from automobile exhaust (de Gouw et al., 2005), as a tracer of primary emissions to roughly estimate contributions of secondary HCHO production.

Actually, we have noticed de Gouw's method to estimate contributions of different sources to ambient HCHO even before the original submission, but we failed to achieve an acceptable fitting. This time,

according to the suggestion, we fully realized the disadvantages of the simple estimation method based on the ratios of HCHO-to-tracers, and spent more time on de Gouw's method again. Unfortunately, the five fitting parameters always fluctuated severely and were influenced intensely by the initial estimates of the parameters, which leads to a large uncertainty of the fitting results and poor correlations of the fitting HCHO and measured HCHO. On the other hand, we found that the multi-linear regression method based on measurements of ambient HCHO and tracers of primary and secondary emissions has been used to separate background, primary, and secondary sources of HCHO in eastern China (where our sampling site is also located), and has been proved to be a valid estimation method (Su et al., 2019; Zhang et al., 2021; Sun et al., 2021). Therefore, we have applied the multi-linear regression method in our revised manuscript to elucidate the sources of HCHO, using $C_2H_2$ as a tracer of primary emissions and $O_3$ as a tracer of secondary production. The fitted HCHO and measured HCHO show a much better correlation (R=0.86) in this method.

We now state in our revised manuscript (Page 8, Line 216-243) that "Here, we present a multi-linear regression method based on measurements on ambient HCHO, $C_2H_2$, and $O_3$ to estimate contributions of background, primary, and secondary HCHO. $C_2H_2$ is used here as an indicator of primary emissions, whereas $O_3$ is used as a tracer of secondary production. A linear regression model was used to establish a link among the time series of HCHO, $C_2H_2$, and $O_3$ (Garcia et al., 2006; Zhang et al., 2021; Su et al., 2019; Sun et al., 2021). The observed HCHO can be reproduced by the following linear regression model as shown in Eq. (1).

$$[HCHO] = \beta_0 + \beta_1[C_2H_2] + \beta_2[O_3] \tag{1}$$

where $\beta_0$, $\beta_1$, and $\beta_2$ are the fitting coefficients calculated from the multiple linear regression, and $[HCHO]$, $[C_2H_2]$ and $[O_3]$ represent the concentrations of HCHO, $C_2H_2$ and $O_3$, respectively. The relative contributions of background, primary, and secondary sources to ambient HCHO can be calculated using Eq. (2)- Eq. (4) (Garcia et al., 2006; Zhang et al., 2021; Su et al., 2019; Sun et al., 2021).

$$R_{Background} = \frac{\beta_0}{\beta_0 + \beta_1[C_2H_2] + \beta_2[O_3]} \tag{2}$$

$$R_{Primary} = \frac{\beta_1[C_2H_2]}{\beta_0 + \beta_1[C_2H_2] + \beta_2[O_3]} \tag{3}$$

$$R_{Secondary} = \frac{\beta_2[O_3]}{\beta_0 + \beta_1[C_2H_2] + \beta_2[O_3]} \tag{4}$$

where $R_{Primary}$ denotes the relative contribution of HCHO from primary sources, $R_{Secondary}$ represents the relative contribution of HCHO from secondary sources, and $R_{Background}$ denotes the relative contribution of background HCHO, which may come from alternated HCHO sources with different residence times or transport of HCHO into the airshed and cannot be classified as primary or secondary sources.

Figure 4 reveals the time series of and percentile contributions of background, primary, and secondary HCHO. The modelled HCHO and measured HCHO show a significant linear regression (R=0.86) (Figure 4a), which confirms that the multiple linear regression model is statistically reliable. Secondary HCHO dominated local HCHO during the daytime, whereas primary HCHO was more important during nighttime and in the early morning, when photochemical reactions could be nearly neglected (Figure 4b). The campaign-average relative contributions of secondary production (Figure 4c) showed the lowest value at 5:00 LT, which was associated with weaker photochemical reactions and higher vehicle emissions due to the morning rush hours. After that, with the increasingly active photochemistry, percentages of secondary HCHO started to rise and reached the peak in the early afternoon (around 13:00-15:00 LT), and then gradually decreased. On average, background, primary, and

secondary HCHO contributed 12.7%, 30.4%, and 56.9% to ambient HCHO during the campaign, whereas during the time period with the most intensive photochemistry (10:00-16:00 LT), their relative contributions were 9.5%, 20.9%, and 69.6%, respectively." The revised Figure 4 is also presented as Figure R1 in this response for the referee's convenience.

Also, a simple estimation method based on the ratios of HCHO-to-tracers was conducted to compare results of the two methods. We now state in our revised supplement (Page 2, Line 16-27) that "A simple method to estimate contributions of primary HCHO and secondary HCHO, based on the ratios of HCHO and C2H2, is shown in Eq. (S1) (Lin et al., 2012).

$$f_{HCHO} = \left(\left[\frac{HCHO}{C_2H_2}\right]_a - \left[\frac{HCHO}{C_2H_2}\right]_{pri}\right)\bigg/\left[\frac{HCHO}{C_2H_2}\right]_a \tag{S1}$$

where $f_{HCHO}$ represents the fraction of secondary HCHO, $\left[\frac{HCHO}{C_2H_2}\right]_a$ represents the HCHO-to-$C_2H_2$ ratios in the ambient air, and the primary ratio $\left[\frac{HCHO}{C_2H_2}\right]_{pri}$, i.e., the initial mixing ratio of HCHO and $C_2H_2$ in the fresh emissions, is assumed to be the $10^{th}$ percentile value (0.73) of all the measured values of HCHO/$C_2H_2$ during our campaign (Wang et al., 2020).

Figure S2 reveals our estimation of average contributions of secondary HCHO during a day. Secondary HCHO exceeded the rest sources including primary and background ones all the time, with the highest values in the afternoon. Secondary HCHO contributed the least to ambient HCHO during 5:00-9:00 LT, whereas at noon and in the afternoon, it showed an obvious enhancement, and then gradually decreased and remained flat during the nighttime. On average, secondary HCHO was estimated to contribute approximately 62.4% to the total ambient HCHO during the campaign, whereas during the time period with the most intensive photochemistry (10:00-16:00 LT), the contribution from secondary HCHO accounted for about 65.7%." The revised Figure S2 is included in this response as Figure R2.

We compare results from the two methods and state in our revised manuscript (Page 10, Line 244-249) that "Another simple method of separating primary and secondary HCHO is shown in the Supplement (Sect. S1) for comparison. Diurnal variations of relative contributions of secondary HCHO from two methods are quite similar, as shown in Figure S2 and Figure 4, though there are tiny differences between the absolute numbers, e.g., 65.7% according to the ratios of HCHO/$C_2H_2$ and 69.6% from the multi-linear regression method, respectively. The estimated contributions of secondary HCHO indicate an important role of secondary HCHO production, especially during the daytime, when the photochemistry is more active."

[Figure]

[Figure]

[Figure]

**Figure R1.** (a) Time series and (b) relative contributions of background, primary, and secondary sources to HCHO from a multi-linear regression model during the campaign, and (c) campaign-average diurnal contributions of background, primary, and secondary sources.

[Figure]

**Figure R2.** Diurnal contributions of secondary HCHO based on emission ratios of HCHO-to-$C_2H_2$, with the error bar representing one standard deviation.

*(2) The calculation of HCHO production is even problematic. On the one hand, Eq. 2 assumes HCHO production has reached steady state, or it represents the maximum HCHO production at given levels of VOC precursors and oxidants. the steady state assumption might be violated when fresh emitted VOCs are mixed in the detected air mass. For example, strong isoprene emission nearby around noon. On the other, Eq. 2 estimates the in-situ HCHO production at the very time point of VOCs measurement. This means, if the detected air mass is to a large extent photochemically aged, the measured HCHO is most likely not formed from the in-situ production. In such a case, results from Eq. 2 cannot represent the real HCHO sources. Therefore, it is better to apply a time-dependent photochemical model for calculating HCHO production.*

**R2:**

Sorry for our vague expression. In fact, Eq. 2 (now Eq. 5 in our revised manuscript) represents calculation of the instantaneous HCHO production flux (at a time interval of 1 hr), which doesn't assume that steady state is reached. Our work aims to investigate whether a steady state is reached, and whether there were important missing production and loss processes of HCHO by comparing the production-loss difference we calculated with the observed rate of change in the HCHO concentration. HCHO production rates were calculated using similar methods shown as Eq. 2 in previous studies (Sumner et al., 2002; Choi

et al., 2010; Lin et al., 2012; Nussbaumer et al., 2021), which have been proved to be a valid way to calculate the local HCHO production flux.

*(3) The authors specifically mentioned in the introduction that the missing production of HCHO exists in many studies. However, they did not describe the situation in this study. Even the authors estimate the contribution of different VOCs to HCHO production, the audience still could not know which measured VOCs / photochemical process / unmeasured species is important for understanding the "missing production".*

**R3:**

According to this suggestion, we now state in our revised manuscript (Page 11, Line 305-311) that "For both the sunny period and the cloudy and rainy period, ethene and propene turned out to be two of the most important precursors, but their significance has not been reported in previous studies, which might be attributed to the forest environments where most previous studies were conducted (Choi et al., 2010; Sumner et al., 2001). Also, the importance of MVK and MACR was not found previously, but these two precursors turned out to play an important role in secondary HCHO production in our study. To the best of our knowledge, there is only one study that has previously calculated HCHO production from various alkanes in an urban area in USA (Lin et al., 2012). Our study takes a wide range of VOCs including alkanes, alkenes, aromatic and OVOCs into the calculation of HCHO production rates.".

*(4) According to Fig. 9a, the calculated net P(HCHO) is generally higher than the measured d[HCHO]/dt during daytime, given the uncertainty of net P(HCHO) (~28%) and HCHO measurement (~10%). This indicates, either P(HCHO) is overestimated or D(HCHO) is underestimated. However, the authors did not go into details of possible reasons.*

**R4:**

We now state in our revised manuscript (Page 14, Line 393-409) that "Any discrepancy between the calculated net HCHO production rates and the observed rates of HCHO concentration change will be due to either unconsidered chemical terms or meteorological effects (Sumner et al., 2001). Figure 10a shows that our calculated net HCHO production rates are in relatively good agreements with the observed rates of HCHO concentration change throughout the day in the sunny period, indicating that our treatment of chemical terms is accurate, and that HCHO was dominantly secondarily produced at the DSL site, which can be approximated by oxidation of the 24 hydrocarbons we considered in our calculation. Thus, transport processes and primary emissions did not significantly impact ambient HCHO concentrations at the DSL site, at least their influences were considerably tiny compared to the chemical production. During nighttime, the rates of HCHO concentration change oscillated around zero, when the calculated production almost completely balanced the loss term. After 7:00 LT, HCHO production exceeded its loss, leading to positive net HCHO production values, which is in line with the increasing trend of the HCHO concentration. After 15:00 LT, HCHO loss began to transcend its production, resulting in the decline of HCHO abundance. Uncertainties of the calculated net HCHO production rates (38.8%) and the observed rates of HCHO concentration change (30%) led to overlapped uncertainty ranges, as shown in Figure 10a. Thus, the calculated net HCHO production rates are very close to the observed rates of HCHO concentration change. However, during 12:00-14:00 LT, the calculated net HCHO production rates were slightly higher than the observed rates of HCHO concentration change by around 0.6 ppbv h$^{-1}$, indicating a missing loss term, most likely due to dilution with HCHO-poor air from the nocturnal residual layer,

which coincided with the obvious enhancements of the boundary layer height that were usually observed during 11:00-14:00 LT at the DSL site, as shown in Figure S7a.", and that (Page 15, line 412-426) "The differences between the calculated net production rates and the observed rates of HCHO concentration change suggest either a missing loss term or an overestimated production. Since differences were still observed from 8:00-13:00 LT in Figure 10b even when uncertainties of the calculated net HCHO production rates (35.7%) and the observed rates of HCHO concentration change (30%) were considered, a real missing loss term was more likely there. Dilution with HCHO-poor air from the nocturnal residual layer due to the increases in the boundary layer height, as we have discussed for the case in the sunny period, might have contributed to the missing loss term, but probably were not the only donor, since the discrepancies found in 12:00-14:00 LT in the sunny days were only half of those found in the cloudy and rainy days, whereas the increases in the boundary layer height in the cloudy and rainy days were smaller compared to those in the sunny days , as shown in Figure S7b. Also, the wind speeds were usually between 1-3 m s$^{-1}$, which indicates that transport effects from areas with lower HCHO concentrations might not be important. There was significantly more rainfall at daytime in the cloudy and rainy days than in the sunny days during our campaign, and thus we consider the other missing loss process might be wet deposition, which has been reported as a dominant one of the total deposition (i.e. dry deposition and wet deposition) during the rainy season (Seyfioglu et al., 2006). Indeed, HCHO is readily soluble in cloud and rainwater with its high Henry's law constant ($\sim 5.5 \times 10^3$ M atm$^{-1}$) and could be efficiently converted to formic acid in warm cloud droplets (Allou et al., 2011; Chebbi & Carlier, 1996; Franco et al., 2021)."

The revised Figure S7 is included in this response as Figure R3.

[Figure]

[Figure]

**Figure R3.** Average diurnal profile of Boundary Layer Height (BLH) in (a) the sunny period and (b) the cloudy and rainy period, respectively, with the error bar representing one standard deviation.

*(5) Line 98: The measured data of HCHO, methane, other trace gases, and meteorological parameters are averaged over 1h. Although the GC system measuring VOCs provides data with 1h time step, the actual sampling time interval might not be 1h. Therefore, the time average should be applied to the VOCs sampling interval.*

**R5:**

    The GC-MS/FID cycle was actually 1 hr as stated in Line 110-112 in Page 4 in our manuscript, which reads, "Measurements of Photochemical Assessment Monitoring Station (PAMS) VOC compounds were performed with GC-MS/FID (TH-PKU 300B) with a time resolution of 1 hour, given the time for heating and cooling of the GC oven (Li et al., 2015).". We believe that our measured VOCs data with 1 hr time resolution are reasonable.

*(6) Line 356: "Figure 9a shows that our net HCHO production are in good agreements with the observed HCHO rate of change". This might not be true as I described above.*

**R6:**

    Please refer to our response to Comment # (2) and Comment # (4) from Reviewer #1.

*(7) Line 362 - 363: The uncertainty of the net P(HCHO) should be given.*

**R7:**

    According to this suggestion, we now state in our revised manuscript (Page 15, Line 390-392) that "Uncertainties of the calculated net HCHO production rates and the observed rates of HCHO concentration change ($\frac{d[HCHO]}{dt}$) are listed in Table S8. The uncertainty of the observed rates of HCHO concentration change is composed of the HCHO measurement uncertainty and the uncertainty of the fit (30% upper limit) (Nussbaumer et al., 2021) with the latter dominating.". Also, Table S8 is also presented as Table R1.

    The uncertainties of the net HCHO production rates and the observed HCHO rates of change are now shown in the revised Figure 10 (also as Figure R4). The figure caption has been revised accordingly, which (Page 35, Line 708-710) reads "Figure 10. Average daytime production rates, loss rates, the calculated net production rates, and the observed rates of HCHO concentration change for (a) the sunny and (b) the cloudy and rainy period. Shaded areas give the uncertainties of the calculated net production rates and the observed rates of HCHO concentration change."

**Table R1.** Uncertainties of the calculated net HCHO production rates and the observed rates of HCHO concentration change during the sunny period and the cloudy and rainy period.

|  | The sunny period (%) | The cloudy and rainy period (%) |
| --- | --- | --- |
| Calculated net HCHO production rates | 38.8 | 35.7 |
| Observed rates of HCHO concentration change | 30 | 30 |

[Figure]

[Figure]

**Figure R4.** Average daytime production rates, loss rates, the calculated net production rates, and the observed rates of HCHO concentration change for (a) the sunny and (b) the cloudy and rainy period. Shaded areas give the uncertainties of the calculated net production rates and the observed rates of HCHO concentration change.

**Reviewer #2** (Formal Review for Author):

*The measurement report on production and loss of atmospheric formaldehyde at a suburban site of Shanghai in summertime describes in-situ HCHO measurements and some of its precursors. Based on these observations HCHO direct emissions, photochemical production and loss, as well as deposition losses are determined. In general, the topic is highly relevant to ACP and the paper is well written. Thus, I recommend publication after some minor revisions.*

**R:**

We appreciate the positive comments from Reviewer #2 and have revised our manuscript accordingly as shown below.

(1) *I think that a number of assumptions have been made for the analysis that could be better motivated. E.g. steady state is generally assumed, although the observations could be used to indicate whether this assumption is met or not. As mentioned in the manuscript, changes in HCHO (dHCHO/dt) can be due to direct emissions, net photochemical production (production – losses), wet and dry deposition and transport (horizontal and vertical). The rate of change can be inferred from the HCHO time line, and production and loss, as well as deposition losses are calculated in the paper. Therefore, any deviations between the rate of change and these production and loss terms are due to either direct emissions or transport. A thorough investigation of the diurnal variation might provide some insight into the dominating mechanisms. E.g. changes in the boundary layer height before noon will lead to downward transport of HCHO poor air from the nocturnal residual layer, yielding a dilution acting as a missing "loss"-term, while traffic related HCHO emissions during the morning and afternoon rush-hours will provide an additional temporal source.*

**R1:**

Please refer to our response to Comment # (4) from Reviewer #1.

(2) *The estimation of HCHO direct emissions based on HCHO vs. NOx relations seems to be an oversimplification. Although it is true that traffic related HCHO emissions are in general associated with NOx emissions, HCHO production is also to some extend NOx dependent (see e.g. Wolfe et al., Mapping hydroxyl variability throughout the global remote troposphere via synthesis of airborne and satellite formaldehyde observations." Proceedings of the National Academy of Sciences, 116 (23): 11171-11180 [10.1073/pnas.1821661116], 2019). In addition, this kind of analysis ignores the HCHO background, which leads to the strange result that according to Fig. 4 50 % of HCHO during the night is due to photochemical sources.*

**R2:**

According to the suggestion, we now use acetylene ($C_2H_2$) instead of $NO_x$ as a tracer of primary emissions. The diurnal variations of HCHO, $O_3$ and $C_2H_2$ at the DSL site during the campaign are now shown in the revised Figure 3a (also as Figure R5a), and correlations between hourly concentrations of HCHO and $O_3$ and those of HCHO and $C_2H_2$ are shown in the revised Figure 3b (also as Figure R5b) and Figure 3c (also as Figure R5c), respectively. We now state in our revised manuscript (Page 8, Line 206-215) that "Figure 3a further illustrates the diurnal variations of HCHO, $O_3$ and $C_2H_2$ at the DSL site during the campaign. Both HCHO and $O_3$ exhibited strong diurnal variations during the field measurement,

which reached the maximum in the early afternoon, decreased gradually in the afternoon, and remained flat at night. The good correlation between hourly concentrations of HCHO and $O_3$ ($R^2$=0.73), as shown in Figure 3b, indicates important contributions of secondary sources to HCHO. By contrast, the variations of HCHO and $C_2H_2$ in Figure 3a are less parallel. $C_2H_2$ increased in the early morning since around 5:00 LT (local time), reached its maximum in the rush hours at 8:00-9:00 LT, then decreased till noon, and remained relatively steady with a much smaller peak in the evening around 20:00 LT, which likely coincided with the traffic volumes on the highroad. The correlation ($R^2$=0.55) between HCHO and $C_2H_2$ was lower compared to the correlation between HCHO and $O_3$, as shown in Figure 3c. These observations roughly indicate that HCHO at the DSL site was more influenced by secondary sources than primary sources."

In addition, a multi-linear regression method has been used instead to estimate the contributions of background, primary, and secondary sources in our revised manuscript. Please refer to our response to Comment # (1) from Reviewer #1.

[Figure]

[Figure]

[Figure]

**Figure R5.** (a) Diurnal patterns of HCHO, $O_3$, and $C_2H_2$, with the error bar indicating one standard deviation; (b) Correlation between measured concentrations of HCHO and $O_3$; (c) Correlation between measured concentrations of HCHO and $C_2H_2$.

(3) *I think it is critical that the precursor measurements miss a number of important oxygenated HCHO precursors like methanol, acetone, acetaldehyde and methylhydrogenperoxide. According to Nussbaumer et al. in particular photolysis of these species yield CH3O2 radicals that at high enough NO levels yield HCHO. The fact that Nussbaumer et al. demonstrated that these species have low mixing ratios at rural/remote sites does not justify it to ignore their potential role in this suburban environement.*

**R3:**

    Indeed, acetone was measured in our campaign. Unfortunately, methanol, acetaldehyde and methylhydrogenperoxide (MHP) were not measured during this campaign.

    Here, we use concentrations of acetaldehyde we measured at the same location in the 2020 summer to estimate the contribution of acetaldehyde to HCHO production, which was $3.00\pm1.13$ ppbv on average. Thus, the contribution of acetaldehyde oxidation by OH radicals to HCHO formation was estimated to be about 3%.

    Also, we searched the literature for measured concentrations of methanol in similar regions to roughly estimate its potential contribution to HCHO production. Measurements of methanol were hardly reported in Shanghai, but measurements in rural areas in southern China and in eastern China (where Shanghai is

located) are available. Previous studies at rural areas reported average concentrations of 3.90 ppbv at Shenzhen Environmental Monitoring Yangmeikeng Station (southern China) in summer (Han et al., 2019), and 5.67 ppbv at a rural site on Changdao Island (eastern China) in late spring (Yuan et al., 2013), respectively. Using these concentrations of methanol and measured concentrations of VOCs in our study, we estimate that methanol of 3.90 ppbv would contribute around 2% to secondary HCHO formation in our study, and methanol of 5.67 ppbv would contribute around 3%.

HCHO yields from acetone and MHP have not been reported in previous studies. To the best of our knowledge, there is only one study that considered HCHO production from acetone and MHP (Nussbaumer et al., 2021), where HCHO production from acetone (or MHP) was estimated using measured photolysis rates of acetone (or MHP), as well as concentrations of NO, OH radicals and $HO_2$ radicals. However, OH radicals and $HO_2$ radicals were not measured during our campaign, and we can't find a valid way to estimate concentrations of $HO_2$ radicals based on our measurement and thus assume a proportional relationship between the HCHO production flux and the acetone/MHP concentration at two locations. The average concentration of acetone in Nussbaumer's study (Nussbaumer et al., 2021) was 2.67 ppbv during the CYPHEX campaign, which was reported to contribute an average daily maximum production rates of around 0.003 ppbv $h^{-1}$. Here, we use concentrations of acetone measured by GC-EI-TOF-MS during our campaign, which showed an average concentration of 6.38 ppbv, to estimate the average daily maximum production rates of about 0.007 ppbv $h^{-1}$, contributing around 0.25% to the HCHO formation at DSL site.

We did not measure MHP during our campaign, and we use the concentration reported in previous studies in China to estimate the potential contribution of MHP to HCHO production. The reported concentrations of MHP were quite similar in different studies, for example, the average measured concentrations of 0.10 ppbv at a suburban site in Beijing (northern China) in summer, and 0.18 ppbv at a rural site in Tai'an city, Shandong Province (eastern China) in summer, respectively (Zhang et al., 2012), the average measured concentrations of 0.13 ppbv at a urban site in Beijing in summer (Qin et al., 2018), and the average measured concentrations of 0.099 ppbv at a urban site in Beijing in autumn (Zhang et al., 2018). The average concentration of MHP in Nussbaumer's study (Nussbaumer et al., 2021) was 0.14 ppbv during the CYPHEX campaign, which was reported to contribute an average daily maximum production rates of around 0.013 ppbv $h^{-1}$. Here, we used an average concentration of 0.18 ppbv for MHP (measured in a suburban site in eastern China (Zhang et al., 2012)) to estimate average daily maximum production rates of around 0.017 ppbv $h^{-1}$, which would contribute around 0.6% to the overall HCHO production.

Therefore, we believe that the precursors we consider almost completely account for the HCHO production at the DSL site. We now state in our revised manuscript (Page 9-10, Line 262-266) that "The VOC species considered are quite comprehensive, though acetaldehyde, methanol, and methylhydrogenperoxide (MHP) are not available in this study. These three compounds, together with acetone, were estimated to contribute up to 7% to secondary HCHO formation at the DSL site, using their concentrations reported in previous studies and that for acetone in our study (Yang et al., 2022; Han et al., 2019; Yuan et al., 2013; Zhang et al., 2012; Nussbaumer et al., 2021). Therefore, we consider their absence would not influence our results considerably."

(4) *Although the determination of dry deposition rates follows the procedure described in Shepson et al. 1992, I am not sure that I agree with some of the conclusions. The determination of deposition losses from nocturnal HCHO mixing ratio changes is only partly due to dry deposition, since in particular in environments with high alkene mixing ratios ozonolysis of these species is a source of HCHO. Thus, the calculated HCHO deposition velocity is most likely a lower limit due to nocturnal HCHO*

*production and not due to neglected turbulence or stomatal up-take. Those processes might be important during the day, yielding higher deposition rates during daylight hours. Nevertheless, a simple multiplication of the deposition rates by a factor 2 as done here is not justified by the approach described by Shepson et al.*

**R4:**

We have considered the impact of ozonolysis of alkenes on the determination of dry deposition velocity. When we chose 9 typical nights to determine the dry deposition velocity of HCHO, we carefully checked concentrations of ozone and alkenes and the corresponding HCHO production rates. The observed HCHO loss rates during these 9 nights were around 0.19 ppbv h$^{-1}$, while the calculated HCHO production rates from ozonolysis of alkenes were about 0.03 ppbv h$^{-1}$, which represents 15% of the dry deposition. In order to be more precise, we now state in our revised manuscript (Page 13, Line 355-357) that "Note that this estimation of the dry deposition loss is a lower limit, since it neglects nighttime production of HCHO due to ozonolysis of alkenes, as well as thermally driven turbulence and deposition caused by stomatal uptake by vegetation (Fischer et al., 2019; Nguyen et al., 2015).", and (Page 13, Line 358-360) that "We have performed this calculation for 9 nights when the estimated HCHO production from ozonolysis of alkenes was around 15% of the observed HCHO loss on average, and an overview of the 9 nights can be found in Figure S5.".

Secondary HCHO production during daytime would bring a large error when determining the HCHO dry deposition velocity from the observed HCHO loss rates. Therefore, dry deposition velocity of HCHO was derived from the nighttime decrease in HCHO. According to Shepson *et al.* (Shepson et al., 1992) and Fischer *et al.* (Fischer et al., 2019), we made a simplified assumption that the height of the boundary layer is constant and assumed a linear increase in the HCHO mixing ratio with height in the nocturnal boundary layer, while during daytime the boundary layer is well mixed and the HCHO mixing ratio was constant at different heights. The factor 2 here was used to consider the different mixing of the boundary layer during nighttime and daytime, not for the influence of photochemical HCHO production.

*(5) Line 20: I think it should read average secondary HCHO net production.*

**R5:**

Sorry for our vague expression. The average secondary HCHO production rate of 0.73 ppbv h$^{-1}$ was derived from the calculated HCHO production rates during the whole campaign, including those in both nighttime and daytime. We now state in our revised manuscript (Page 1, Line 21-24) that "Average secondary HCHO production rate was estimated to be 0.73 ppbv h$^{-1}$ during the whole campaign (including daytime and nighttime), with a dominant contribution from reactions between alkenes and OH radicals (66.3%), followed by OH radical-initiated reactions with alkanes and aromatics (together 19.0%), OH radical-initiated reactions with OVOCs (8.7%), and ozonolysis of alkenes (6.0%)."

*(6) Line 96: To my knowledge, the Aerolaser instrument uses an UV-LED at 410 nm and not laser at 400 nm.*

**R6:**

Sorry for the typo. We have checked the principle of the Aerolaser instrument. It's true that 3,5-diacetyl-1,4-dihydrolutidine (DDL) absorbs light at 410 nm. We have corrected this mistake in our revised manuscript (Page 4, Line 98-101) that "In this instrument, gaseous HCHO is firstly absorbed by sulfuric acid (0.3%) and transferred into liquid phase, and then the aqueous HCHO reacts with acetylacetone and

ammonia to produce 3,5-diacetyl-1,4-dihydrolutidine (DDL), which can be excited by a laser at 410 nm and the fluorescence at 510 nm is used to quantify HCHO.".

(7) *Line 160: Do you consider only downward welling radiation or also up-ward welling radiation for the calculation of photolysis rates?*

**R7:**

The photolysis frequencies measured by the Ultra-fast CCD-Detector Spectrometer (Metcon, UF-CCD) only consider downward welling radiation and thus our calculation of photolysis rates only involves downward welling radiation.

**References**

[revised manuscript text omitted]

---

## Author Response (AR2)

**A point-to-point response to reviewers' comments**

We are very grateful to the helpful and insightful comments from Reviewer #1, and have carefully revised our manuscript accordingly. In the following point-to-point response, Reviewer #1's comments are repeated in *italics* whereas our responses are in plain texts labelled with **[R]**. Line numbers in the responses correspond to those in the revised manuscript (the version with all changes accepted). Modifications to the manuscript are in blue.

**Reviewer #1** (Formal Review for Author):
*The authors have well addressed most of my concerns, and the manuscript has been revised substantially. I recommend the publication as it is unless the editor thinks my following comment needs to be well considered.*

*I don't agree with the argument about the time resolution of the GC-FID/MS system, i.e., R5. The authors should pay attention that the the system (i.e., TH-300B) usually takes air samples for 10 - 30 min and then performs the GC-FID/MS analysis. Although one analysis cycle takes 1 hour, the air sample been detected is not the one sampled in the entire one hour. Therefore, when perform time average or time synchronization, the time period been considered should refer to the exact time period when the TH-300B takes air samples.*

**R:**
We appreciate the positive comments from Reviewer #1 and have revised our manuscript accordingly as shown below.

Indeed, the air sample detected by the GC-MS/FID system did not last the entire one hour, which may miss spikes of high/low concentrations of VOCs. However, for a long-time campaign (that lasted for 600 hours), the air sample taken in each sampling cycle can represent the general chemical composition and trend of the ambient air. We now state in our revised manuscript (Page 4, Line 110-115) that "Sampling of Photochemical Assessment Monitoring Station (PAMS) VOC compounds were performed for 10 min each hour, and the resulting data from GC-MS/FID (TH-PKU 300B) analysis were used to represent concentrations of PAMS compounds in that hour, given the time for heating and cooling of the GC oven (Li et al., 2015). This practice may miss spikes of PAMS concentration variation, but on a longer time scale, the general characteristics of ambient air composition and concentrations likely still show similar features in one hour (Kumar and Sinha, 2014; Li et al., 2015; Yuan et al., 2012; Yang et al., 2022)."

**References**

Kumar, V. and Sinha, V.: VOC–OHM: A new technique for rapid measurements of ambient total OH reactivity and volatile organic compounds using a single proton transfer reaction mass spectrometer, Int. J. Mass Spectrom., 374, 55–63, https://doi.org/https://doi.org/10.1016/j.ijms.2014.10.012, 2014.

Li, J., Xie, S. D., Zeng, L. M., Li, L. Y., Li, Y. Q., and Wu, R. R.: Characterization of ambient volatile organic compounds and their sources in Beijing, before, during, and after Asia-Pacific Economic Cooperation China 2014, Atmos. Chem. Phys., 15, 7945–7959, https://doi.org/10.5194/acp-15-7945-2015, 2015.

Yang, G., Huo, J., Wang, L. L., Wang, Y., Wu, S., Yao, L., Fu, Q., and Wang, L. L.: Total OH Reactivity Measurements in a Suburban Site of Shanghai, J. Geophys. Res. Atmos., 127, e2021JD035981, https://doi.org/https://doi.org/10.1029/2021JD035981, 2022.

Yuan, B., Shao, M., De Gouw, J., Parrish, D. D., Lu, S., Wang, M., Zeng, L., Zhang, Q., Song, Y., Zhang, J., and Hu, M.: Volatile organic compounds (VOCs) in urban air: How chemistry affects the interpretation of positive matrix factorization (PMF) analysis, J. Geophys. Res. Atmos., 117, 1–17, https://doi.org/10.1029/2012JD018236, 2012.